

# Estimation of Tropical Cyclone Wind Hazards in Coastal Regions of China

Genshen Fang[1,3], Lin Zhao[1,2], Shuyang Cao[1,2], Ledong Zhu[1,2], Yaojun Ge[1,2]

[1]State Key Lab of Disaster Reduction in Civil Engineering, Tongji University, Shanghai 200092, China;

[2]Key Laboratory of Transport Industry of Wind Resistant Technology for Bridge Structures, Tongji University, Shanghai 200092, China;

[3]Glenn Department of Civil Engineering, Clemson University, Clemson, SC 29634, USA

*Correspondence to*: Lin Zhao (zhaolin@tongji.edu.cn)

**Abstract.** Coastal regions of China feature high population densities as well as wind-sensitive structures and are therefore

vulnerable to tropical cyclones (TCs) with approximately 6~8 landfalls annually. This study predicts TC wind hazard curves in terms of design wind speed versus return periods for major coastal cities of China to facilitate TC-wind-resistant design and disaster mitigation as well as insurance-related risk assessment. 10-min wind information provided by the Japan Meteorological Agency (JMA) from 1977 to 2015 is employed to rebuild TC wind field parameters (radius to maximum winds $R_{max,s}$ and shape parameter of radial pressure profile $B_s$) at surface level using a height-resolving boundary layer model. These

parameters will be documented to develop an improved JMA dataset. The probabilistic behaviours of historical tracks and wind field parameters at the first time step within a 500-km-radius subregion centred at a site of interest are examined to determine preferable probability distribution models before stochastically generating correlated genesis parameters utilizing the Cholesky decomposition method. Recursive models are applied for translation speed, $R_{max,s}$ and $B_s$ during the TC track and wind field simulations. Site-specific TC wind hazards are studied using 10,000-year Monte Carlo simulations and

compared with code suggestions as well as other studies. The resulting estimated wind speeds for northern cities (Ningbo and Wenzhou) under TC climates climate are higher than code recommendations while those for southern cities (Zhanjiang and Haikou) are lower. Other cities show a satisfactory agreement with code provisions at the height of 10 m. Some potential reasons for these findings are discussed to emphasize the importance of independently developing hazard curves of TC winds under non-synoptic climates.

# 1 Introduction

Tropical cyclones (TCs) are rapidly rotating storms characterized by strong winds, heavy rain, high storm surges and even devastating tornadoes. They inflict tremendous damage on property and considerable loss of human life and pose threats to flexible structures in coastal areas (Done et al., 2019). In the Western Pacific Basin, TCs form throughout the year. It is the most active TC basin in the world, producing more than 30 storms annually, accounting for almost one-third of the global total

(Knapp et al., 2010; Yang and Chen, 2019). The Southeast China coastal area has long coastlines and numerous islands, which



is featured with high population densities as well as many wind-sensitive structures including high-rise buildings and long-span bridges. It is a TC-prone region, with an average of 6~8 TC landfalls per year. It has been estimated that more than 1,600 fatalities and 80 billion RMB of direct economic loss can be attributed to TCs and subsequent floods in 2006 alone in coastal regions of China (Liu et al., 2009), demonstrating that this area is extremely vulnerable to TC damage. Accordingly, it is an

issue of great importance to analyse TC wind hazards to support wind-resistant design as well as disaster mitigation and insurance-related risk assessment.

Unlike synoptic winds such as monsoons, TCs are moving rotating storms with a small occurrence rate at a specific location. Moreover, wind anemometers are usually vulnerable to damage during strong typhoon events, making the record of historically observed winds an unreliable predictor for design wind speed based on statistical distribution models. The largest yearly wind

speed dataset derived from both synoptic and TC winds is considered to be not well-behaved because the contribution of each wind speed to describe the probabilistic behaviour of the extreme winds is inhomogeneous (Simiu and Scanlan, 1996). An alternative approach, called stochastic simulation or Monte Carlo simulation, introduced in the 1970s by some pioneering studies (e.g. Russell and Schueller, 1971; Batts et al., 1980), has been widely adopted to stochastically generate a large number of wind speed samples using historical data-based probability distributions of several key field parameters. In order to achieve

TC-hazard assessment by Monte Carlo simulation, the circular sub-region method (CSM) was developed by Georgiou (1985) and later employed by Vickery and Twisdale (1995), Xiao et al. (2011) and Li and Hong (2015). CSM uses the circled historical track information centred on the site of interest to characterize the statistics of some TC parameters before conducting storm simulation and wind speed prediction. This is a site-specific approach. The state-of-the-art empirical full track technique was first developed by Vickery et al. (2000b) and followed by FEMA (2015) as well as ASCE 7-16 loads standard (2017) and Li

et al. (2016), which simulate the TC tracks as well as the intensity in terms of a relative intensity index from genesis to lysis, facilitating the TC risk assessments for the whole coastal region. Although the full track model is preferable for modelling the TC hazards along the whole coastline, CSM is widely used for some site-specific TC risk studies and can be easily updated and improved by supplementary observations. This is also adopted in this study.

During TC wind estimation, the parametric TC wind field model has been commonly adopted and has been continuously

improved over the past several decades based on the ever-increasing amount of observation data. This model is considered to be more economical with time and even more accurate in predicting TC wind velocity compared with some meteorological models. Some pioneering studies on parametric TC wind field modelling have been performed since the 1980s (Batts et al., 1980; Georgiou, 1985; Vickery et al., 2000a, 2009; Nederhoff et al., 2019; Arthur, 2019). These studies employed a gradient wind speed model solved by the atmospheric balance equation of a stationary storm coupled with a depth-averaged (Vickery

et al., 2000a) or a semi-empirical observation-based boundary layer vertical profile model (Vickery et al., 2009). In recent years, with advances in computing capacity, another more sophisticated physical model has received intensive attention. This is the so-called height-resolving model, in which the boundary layer wind field is solved semi-analytically based on 3D Navier-Stokes equations (Meng et al., 1995; Kepert, 2010; Snaiki et al., 2017; Fang et al., 2018). This is of great help in interpreting the underlying physics of the TC boundary layer.



Conventionally, wind field parameters such as the radius to maximum wind speed $R_{max}$ and shape parameter of radial pressure
       profile $B$ were statistically modelled as functions of surface central pressure deficit, TC eye centre latitude and sea surface
       temperature (Vickery et al., 2000b, 2008; Xiao et al., 2011; Zhao et al., 2013; FEMA, 2015; Fang et al., 2018). This facilitated
       TC-related hazard assessment by carrying out a large number of scenarios using Monte Carlo algorithm since the historical
       track information is readily available in each best-track dataset. However, the correlations between these parameters were not

very strong, as shown by Vickery et al. (2000b), with all coefficients of determination less than 0.30. The auto correlations of
       $R_{max}$ as well as $B$ between different time steps in these studies were usually propagated from surface pressure deficit and sea
       surface temperature, which were integrated with a term of relative intensity and modelled by a recursive model. Moreover, the
       cross-adoption of these parameter models in different basins could cause some undesired results since they are always region-
       dependent due to differences among macroscopic atmospheric thermodynamic environments.

In this study, wind field information of 10 min time duration provided by the best track dataset of the Japan Meteorological
       Agency (JMA) was adopted to develop a dataset of $R_{max}$ and $B$ at surface level ($R_{max,s}$ and $B_s$) using a height-resolving wind
       field model. Then the TC design wind speed was predicted by following the procedures illustrated in Fig.1. Based on the
       historical track information extracted from the JMA dataset within a circular subregion with a radius of 500 km centred at the
       site of interest, the preferable probabilistic distributions of six genesis parameters at the first time step, the position of the first

track dot ($\alpha_0$), heading direction ($\theta_{T0}$), central pressure difference ($\Delta P_0$), translation speed ($V_{T0}$), $R_{max,s0}$ and $B_{s0}$ would be
       determined before performing the correlation analyses. Site-specific recursive models of translation speed as well as $R_{max,s0}$
       and $B_{s0}$ were developed using the track information within the circular subregion. Finally, 10,000-year Monte Carlo
       simulations were conducted to investigate the TC wind hazard for 10 coastal cities of China.

## 2 Statistical characteristics of TC tracks

### 2.1 JMA best track dataset

       In the Western Pacific Basin (0°~60°N, 100°~180°N), the Japan Meteorological Agency (JMA) serves as the Regional
       Specified Meteorological Center (RSMC, 2018), as specified by the World Meteorological Organization (WMO). As such, it
       is responsible for forecasting, naming, tracking, distributing warnings and issuing advisories of TCs. Accordingly, JMA has
       been publicly releasing best track datasets of TCs in the Western Pacific Basin since 1951. These datasets contain not only

some basic track information of TCs in terms of latitude and longitude of TC eye centres as well as dates and times, but also
       some wind speed information including minimum surface central pressure ($P_{cs}$), maximum sustained surface wind speed
       ($V_{max,s}$) and 50-knot or 30-knot winds radii estimated from surface observation, ASCAT observation and low-level cloud
       motion satellite images. Although some other organizations issue their own track dataset of TCs for the Western Pacific Basin
       (Ying et al., 2014), such as the China Meteorological Administration (CMA), Joint Typhoon Warning Center (JTWC), the

Hong Kong Observatory (HKO) and the International Best Track Archive for Climate Stewardship (IBTrACS) project, there





are some inconsistencies among these datasets that should be carefully considered. In addition to differences of TC track information and annual TC frequencies, two typical TC intensity representations, i.e. $P_{cs}$ and $V_{max,s}$, show inconsistency from agency to agency, as discussed by Song et al. (2010). Generally, a remarkable difference was found, i.e., that $V_{max,s}$(JTWC) > $V_{max,s}$(CMA) > $V_{max,s}$(JMA) and $P_c$(JTWC) < $P_c$(CMA) < $P_c$(JMA), when TCs reach typhoon level, and this trend becomes apparent along with storm intensification (Song et al. 2010). It could attribute to time interval differences since JMA uses 10 min, CMA uses 2 min while JTWC uses 1 min is adopted by JTWC. The differences among estimation techniques and algorithms for determining $V_{max,s}$ and $P_{cs}$ based on the Dvorak technique (Dvorak, 1984; Velden et al., 2006) with satellite cloud images could also contribute to this inconsistency. However, the 10-min time duration employed by JMA is consistent with most design codes or standards, and is also suggested by WMO (Fang et al., 2019). Furthermore, the 50-knot or 30-knot radii information provided by the JMA dataset is a supplement of great importance in facilitating the estimation of TC wind field parameters. As a result, the JMA best track dataset was selected as the basic information for the following TC hazards studies in the Southeast China region.

## 2.2 JMA best track dataset

In order to examine the statistical characteristics of historical track information around a site of interest, track segments that intersect and are within a circular sub-region entered at the target location are usually extracted from the best track dataset. The size of the subregion directly affects the data sampling as well as final design wind speed prediction (Georgiou, 1985; Xiao et al., 2011; Li and Hong, 2015). A suitable circle size should enable the TC tracks and wind field parameters to be least sensitive and to cover as many high wind speed samples as possible. Three radii, 500 km, 1000 km and 250 km were employed by Vickery and Twisdale (1995), Xiao et al. (2011) and Li and Hong (2015), respectively. The use of 1000 km could overestimate the effects of high winds on a site of interest since some extremely violent typhoons over distant sea would be circled and used to model the central pressure before landfall. However, these typhoons have little chance of maintaining an extremely high intensity until landfall on mainland China. Based on the JMA dataset from 1951 to 2015, only seven violent typhoons ($P_{cs} \leq 935\ hPa$ or $V_{max,s} \geq 54\ \mathrm{m/s}\ (105\ \mathrm{knots})$), Nina (195307), Wanda (195606), Grace (195819), Saomai (200608), Hagupit (200814), Usagi (201319) and Rammasun (2014) directly landed on mainland China. Moreover, the largest $R_{max,s0}$, illustrated in Figs. 8 and 16, range from 500 km to 600 km if the size of subregion R = 500 km is employed. And as mentioned by Yuan et al. (2007), about 50% of the radii of historical storms associated with a wind speed of 15.4 m/s range from 222 km to 463 km and only 10% are larger than 555 km. In fact, we can show experimentally that at the outer regions of a typhoon, 500 km or larger away from storm center would have only a slight influence on the specific region. Accordingly, R = 500 km, which is consistent with Vickery and Twisdale (1995) and will be used in this study, allows as many high wind speeds as possible to be considered and avoids the overuse of some extremely violent typhoons.

Taking the example of the Hong Kong region (centred in 114.1678°E, 22.3186°N), which is severely affected by TCs, 412 segments of track data within a circle of R =500 km were captured from the JMA dataset (1951-2015), as shown in Fig. 2.



Although few TCs originate in this circular region, they only reach the strongest level of a severe tropical storm with $P_{cs}$ larger

than 980 hPa belonging to a normal-intensity storm. Their genesis locations are also close to the circular boundary. Accordingly,

all simulated tracks can be assumed to originate from the circular boundary by considering the location distribution of historical

tracks in term of origin angle $\alpha_0$, which is the direction relative to the site of interest and clockwise positive from the north.

The annual storm rate (storms/year) is usually modelled by negative binomial (Li et al., 2016) or Poisson distributions (Xiao

et al., 2011; Li and Hong., 2015). However, the mean of the storm genesis within the circular region around Hong Kong is

6.339, which is larger than the variance of 2.280. It does not satisfy the prerequisite of the negative binomial distribution. The

Poisson distribution was employed to model the annual storm rate ($\lambda_a$), as shown in Fig. 3. Based on the circular sub-region

method, the position of first track dot ($\alpha_0$) and its heading direction ($\theta_{T0}$) determines the location of the simulated track line

while the translation speed ($V_{T0}$) is used to estimate the TC center location at each time step. First values of the central pressure

difference ($\Delta P_0$) for each segment are applied for the TC intensity modelling before landfall. Based on the statistical

characteristics of historical data, the probabilistic distributions of these four parameters are fitted with several commonly used

models using a maximum likelihood method before achieving the most suitable choices by the K-S distribution test. The

preferable distribution models, i.e. Weibull, lognormal, bimodal normal and Burr type XII for all genesis parameters and their

probability density functions (PDF) together with fitted coefficients are listed in Table 1. Correspondingly, Fig.4 compares the

observed and modelled cumulative distribution functions (CDF) for these parameters. The critical value of K-S test for the

historical data sample (n = 204) is 0.0952 at a 5% significance level larger than all the modelled results (values of k in Fig.4),

which proves that we have enough evidence to simulate the virtual TC tracks by adopting these distribution models. It

noteworthy that all observed $\Delta P$ and $\theta_T$ within the circle of interest are employed to model the distribution of $\Delta P_0$ and $\theta_{T0}$ due

to the inherent drawback of the circular sub-region method, which assumes for simplicity in the simulation that $\Delta P$ remains

unchanged before the storm's landfall and $\theta_T$ is a constant for each TC track. All information of $\Delta P$ and $\theta_T$ can be taken into

account to some extent when they are applied for modeling the distribution of $\Delta P_0$ and $\theta_{T0}$.

**2.3 Translation speed**

The translation speed is used for determining the TC eye locations at every time step and contributes slightly to the TC wind

speed field. Traditionally, it was randomly sampled from a historical-data-based probability distribution (Xiao et al., 2011; Li

and Hong, 2015). In reality, the translation speed of the next step should be correlated with previous steps which is also the

statistical basis for empirical full track modelling (Vickery et al., 2000b; Li et al., 2016). As the real data (historical

observations) illustrated in Fig. 6a~c, the TC translation speed in the Hong Kong region is strongly dependent on the previous

two steps with correlation coefficients of 0.7729 and 0.6281, while a weak correlation is observed with the heading angles.

Accordingly, given the initial storm forward speed, the new speed for next steps can be modelled as a recursive formula

$$lnV_T(i+1) = v_1 + v_2 \cdot lnV_T(i) + v_3 \cdot lnV_T(i-1) + v_4 \cdot \theta_T(i+1) + \varepsilon_{lnV_T} , \tag{1}$$



in which $v_j(j = 1\sim4)$ are model coefficients obtained from the least squares regression analysis for historical data, $V_T(i)$ is

the translation speed at time step $i$, $\varepsilon_{lnV_T}$ is the error term accounting for modelling differences between the regression models

and the real observations.

Based on the JMA dataset, the values of $v_j(j = 1\sim4)$ are extracted as 0.3089, 0.6338, 0.1504 and 0.0001 for the circular Hong

Kong region. Model errors, as illustrated in Fig. 5a, are randomly distributed with mean and standard deviation of 0 and 0.38,

respectively, which indicates that the model is unbiased and has no obvious trend. These errors are then statistically fitted with

two types of probability distribution models, i.e. normal distribution and t location-scale distribution, which are formulated by

the PDFs as

$$f(x;\mu,\sigma) = \frac{1}{\sigma\sqrt{2\pi}}exp\left\{\frac{-(x-\mu)^2}{2\sigma^2}\right\}, \tag{2}$$

$$f(x;\mu,\sigma,\nu) = \frac{\Gamma\left(\frac{\nu+1}{2}\right)}{\sigma\sqrt{\nu\pi}\Gamma\left(\frac{\nu}{2}\right)}\left[\frac{\nu+\left(\frac{x-\mu}{\sigma}\right)^2}{\nu}\right]^{-\frac{\nu+1}{2}}, \tag{3}$$

in which $\mu, \sigma$ and $\nu$ are location, scale and shape parameters. $\Gamma(\cdot)$ is the Gamma function. As shown in Fig. 5b, the normal and

t location-scale distributions are separately applied for to fit the model errors using the maximum likelihood method. Although

the fitting results for both distributions look good, the critical value of the K-S test for the observation data sample (n = 1060)

is 0.0418 at the 5% significance level, which is smaller than the K-S value fitted by normal distribution ($\mu = 0, \sigma = 0.38$) but

larger than that of t location-scale distribution ($\mu = 0.0105, \sigma = 0.2686, \nu = 3.5871$). Consequently, t location-scale

distribution is the preferable distribution for this case and will be used for error sampling.

As shown in Fig. 6, the forward speeds for next steps are modelled by Eq. (1) by introducing the historical track information

and compared with observations. The first row (Fig. 6a~c) only considers the mean terms of Eq. (1), which indicates that the

forward speed significantly depends on the previous steps using the linearly concentrated modelled mean values. The modelled

mean values are more scattered with variation of translation speeds at the previous second step and heading directions, but

they are still within the scatter range of historical data. The second row, i.e. Fig. 6d~f, introduces the error term $\left(\varepsilon_{lnV_T}\right)$

modeled by t location scale distribution (Eq. (3)) as mentioned before, which shows good agreement with the JMA observations.

That is, the translation wind speeds can be well generated using the recursive model of Eq. (1).

## 3 Wind field model

### 3.1 TC wind field solutions

A height-resolving TC boundary layer model developed by Meng et al. (1995) and enhanced by Fang et al. (2018) is adopted

in this study. It is also used to extract two typical TC wind field parameters: radius to maximum wind speed ($R_{max,s}$) and radial

pressure profile shape parameter ($B_s$) at surface level. It is then used to estimate the TC wind speed. Like most parametric TC





wind field models, the surface pressure distribution in the radial direction is always prescribed and formulated by the Holland (1980) model, which is empirically determined by the location parameter ($R_{max,s}$) and the shape parameter ($B_s$) to solve the air pressure term in the Navier-Stokes equation. By extending the Holland pressure model in the vertical direction using the

gas state equation, accounting for the effects of temperature and moisture, a height-resolving parametric TC pressure field model is developed as (Fang et al., 2018)

$$P_{rz} = \left\{ P_{cs} + \Delta P_s \cdot exp\left[ -\left(\frac{R_{max,s}}{r}\right)^{B_s}\right]\right\} \cdot \left(1 - \frac{gkz}{R_d\theta_v}\right)^{\frac{1}{k}},    \tag{4}$$

in which subscripts $r, z$ and s denote values at radius $r$, height $z$ and surface (nominal height 10 m.), respectively. $P_{rz}$ = air pressure at height $z$ and radius $r$ from the TC's axis (hPa), $P_{cs}$ = surface central pressure (hPa), $\Delta P_s = P_{ns} - P_{cs}$ is the central

pressure difference (hPa), where $P_{ns}$ is the peripheral pressure (usually taken as the pressure associated with the outermost closed isobar, 1013hPa in this study), $g = 9.8 N/kg$ is gravitational acceleration, $R_d = 287 J/(kg \cdot K)$ is the specific gas constant of dry air, $\theta_v$ = virtual potential temperature (K), and $k = R/c_p$ is the ratio of gas constant of moist air ($R$) and to specific heat at constant pressure ($c_p$). After that, the wind speed in free atmospheric air can be readily solved. The wind field solutions in the TC boundary layer based on the linearization of Navier-Stokes equations can be expressed as the sum of

gradient wind speed $(V_g)$ and decay wind speeds $(u_d, v_d)$ due to frictional effects. More details regarding the wind field solutions are available in Fang et al. (2018), which are omitted herein for brevity. Some improvements are that the mixing length for determining the eddy viscosity is no longer a linear equation with height, but an upper bound $l_\infty$ of 1/3 boundary layer depth is introduced as suggested by Apsley (1995). That is, the mixing length is modelled as

$$l_v = \left[\frac{1}{\kappa(z+z_0)} + \frac{1}{l_\infty}\right]^{-1},    \tag{5}$$

in which $z_0$ is the equivalent roughness length (m), $\kappa \approx 0.4$ is the von Kármán constant.

**3.2 Wind field parameters**

Two typical parameters, $R_{max,s}$ and $B_s$, are always predefined to model the surface pressure field before solving the wind speed. The JMA best track dataset is a preferable option for TC hazard assessments in the Western Pacific. Its wind speed information in terms of maximum sustained surface wind speed ($V_{max,s}$) and 50-knot or 30-knot winds radii is of great help in

extracting $R_{max,s}$ and $B_s$. Although JTWC also provides information of $V_{max,s}$ as well as the wind radii with respect to 34 knot, 50 knot and 64 knot and radius of maximum winds, the time-averaging issue should be carefully taken into account. Moreover, this wind information in the JTWC dataset is only available from 2001 while JMA documents extend over a longer record from 1977, so is more reliable for developing the parent distribution for use in Monte Carlo simulation. Accordingly, $R_{max,s}$ and $B_s$ used in this study were extracted from the JMA best track dataset (from 1977 to present) by using 50-knot- or

30-knot-radii information as well as the maximum sustained surface wind speeds. These wind data are applied to the



aforementioned wind speed model to derive optimal pairs of $R_{max,s}$ and $B_s$ by minimizing errors between model and observations. For example, in Fig. 7, three radial wind profiles modelled by the optimally fitted $R_{max,s}$ and $B_s$ closely match the JMA observation winds. It is noteworthy that the fitted values of $B_s$ are slightly higher than traditional results, i.e. Vickery et al. (2000b, 2008) while $R_{max,s}$ are almost unchanged. This is mainly attributed to the wind field model used in this study, which transfers the surface pressure field to the gradient layer before working out the surface wind speed using a height-resolving boundary model. As a result, a higher $B_s$ needs to be employed to achieve a strong enough gradient wind field before it is converted to surface level.

Then, the values of $R_{max,s0}$ and $B_{s0}$ associated with the track genesis are determined from their probability distributions considering correlations with other parameters. As shown in Fig. 8, $R_{max,s0}$ and $B_{s0}$ are modelled by lognormal $(\mu = 4.822; \sigma = 0.571)$ and Burr type XII $(\alpha = 1.974, c = 6.362, k = 2.001)$ distributions, respectively. The critical value of K-S test (n = 161) is 0.1059 at a 5% significance level larger than the test statistics (k values in Fig. 8), which fails to reject the null hypothesis. Their correlations with other parameters are also introduced and discussed in the next section.

By using the fitted results from the JMA dataset, the autocorrelations of $R_{max,s}$ as well as $B_s$ between different time steps are simply taken into account using the recursive models as

$$lnR_{max,s}(i+1) = r_1 + r_2 \cdot lnR_{max,s}(i) + r_3 \cdot lnR_{max,s}(i-1) + r_4 \cdot \Delta P_s(i+1) + \varepsilon_{lnR_{max,s}} , \tag{6}$$

$$B_s(i+1) = b_1 + b_2 \cdot B_s(i) + b_3 \cdot B_s(i-1) + b_4 \cdot lnR_{max,s}(i+1) + b_5 \cdot \Delta P_s(i+1) + \varepsilon_{B_s} , \tag{7}$$

in which $r_j (j = 1{\sim}4)$ and $b_j (j = 1{\sim}5)$ are model coefficients that can be fitted with the least squares regression method, $lnR_{max,s}(i)$ and $B_s(i)$ are values at time step $i$, and $\varepsilon_{lnR_{max}}$ and $\varepsilon_{B_s}$ are error terms accounting for modelling differences between the models and observations. Using the data within the Hong Kong region from 1977 to 2015, the values of $r_j (j = 1{\sim}4)$ and $b_j (j = 1{\sim}5)$ are extracted as 0.7039, 0.8341, 0.0282, -0.0016 and -0.6647, 0.5432, -0.0112, 0.2950, 0.0013. As illustrated in Fig. 9a,c, there is no obvious bias or potential trend for the error terms of $lnR_{max,s}$ and $B_s$ with mean $(\mu)$ and standard deviation $(\sigma)$ of 0, 0 and 0.29, 0.20, respectively. Like the translation speed modelled in section 2.3, the error terms of $lnR_{max,s}$ and $B_s$ are both fitted with normal and t location-scale distributions (Fig. 9b, d). It can be noted that both distributions are good candidates for reconstructing the errors, but t location-scale distribution performs better with smaller K-S values (0.029 and 0.028 for $\varepsilon_{lnR_{max,s}}$ and $\varepsilon_{B_s}$) while the critical value of the K-S test for the observation data sample (n = 799) is 0.0478 at a 5% significance level. The fitted parameters for $\varepsilon_{lnR_{max,s}}$ and $\varepsilon_{B_s}$ with t location-scale distribution are $\mu = 0.0107, \sigma = 0.1470, \nu = 2.0340$ and $\mu = 0.0054, \sigma = 0.1461, \nu = 4.1558$, respectively.

As shown in Figs. 10~11, $R_{max,s0}$ and $B_{s0}$ modelled by Eq. (6) and Eq. (7) using the JMA historical data for previous steps are also compared with real observations for next steps. Similarly, the first rows in these two figures ignore the error terms, which are taken into account in the second rows. The values of previous first steps are observed to dominate the model results





with linearly concentrated predictions while previous second steps and other parameters have weaker effects with more scattered model values. After introducing the error terms, model values are able to successfully capture the historical data.

### 3.3 Decay model

Once the storm makes landfall, the central pressure deficit will witness a sudden decrease due to the cut-off of warm and moist
air from the underlying oceanic environment, after which the TC intensity decay model or filling-rate model is adopted. The modelling of storm decay is of great importance for accurately estimating the TC design wind speed at the site of interest since the maximum winds normally occur during storm landfall in most cases. Georgious (1985) modelled the decay of central pressure as a function of distance after landfall for four regions of the United States based on historical data. The other commonly used filling-rate model assumes that the central pressure deficit decays exponentially with time after landfall in the
form of (Vickery, 2005)

$$\Delta P(t) = \Delta P_0 \cdot exp(-at) , \tag{8}$$

in which $t$ is the time after landfall (hour), $\Delta P_0$ is the central pressure difference at landfall (hPa), and $a$ is called the decay rate, which is correlated with $\Delta P_0$ and modelled as

$$a = a_1 + a_2 \Delta P_0 + \varepsilon_a , \tag{9}$$

where $a_1$ and $a_2$ are two region- and topographic-dependent coefficients, and $\varepsilon_a$ is a zero-mean normally distributed error term. As shown in Fig. 12a, the decay information of the ratio of central pressure deficit was extracted from the landfall TCs in the circular region around Hong Kong (Fig. 2) and fitted with the decay model of Eq. (8) using a least squares analysis. Generally, the decay model is well-behaved although it is unable to capture the unchanged central pressures with time after landfall. This is also discussed in detail by Vickery (2005). Furthermore, the correlation between decay rate and central pressure
difference at landfall is plotted in Fig. 12b with the correlation coefficient $\rho = 0.3019$, which is also modelled by the linear function of Eq. (9). Then the residual error is unbiased and can be modelled by a normal distribution with mean and standard deviation of 0 and 0.0227, respectively.

### 4 TC design wind speed prediction

### 4.1 Parameter correlations

As shown by the scatter plots in Fig. 13, the observed (red triangles) genesis (at first time step) parameters show some correlations, especially between $\theta_0$ and $\alpha_0$, $R_{max,s0}$ and $B_{s0}$ with correlation coefficients larger than 0.5. This means that the heading direction at the first time step is dependent on genesis location and two wind field parameters are strongly correlated with each other. Accordingly, the correlations between these genesis parameters, i.e. $\alpha_0$, $\Delta P_0$, $\theta_0$, $V_{T0}$, $R_{max,s0}$ and $B_{s0}$, would be considered when utilizing the Cholesky decomposition method, which is a distribution-free approach introduced by Iman




and Conover (1982). The randomly generated independent variables can be written into a matrix of size N×6 (N is the number

of simulation samples) as

$$\mathbf{X} = \left[ \boldsymbol{\alpha_0}, \Delta \boldsymbol{P_0}, \boldsymbol{\theta_0}, \boldsymbol{V_{T0}}, \boldsymbol{R_{max,s0}}, \boldsymbol{B_{s0}} \right],\qquad(10)$$

The correlation coefficient matrix is $\mathbf{C}$ and is derived from historical data of size 6×6, which is positive definite and symmetric

and can be alternatively expressed as $\mathbf{C} = \mathbf{A}\mathbf{A^T}$ using the Cholesky decomposition method, in which $\mathbf{A}$ is a lower triangular

matrix. If the correlation matrix of $\mathbf{X}$ is $\mathbf{Q}$, it can also be decomposed into the product of a lower triangular matrix $\mathbf{P}$ and its

transpose $\mathbf{P^T}$, i.e. $\mathbf{Q} = \mathbf{P}\mathbf{P^T}$. A matrix $\mathbf{S} = \mathbf{A}\mathbf{P^{-1}}$ can be determined such that $\mathbf{S}\mathbf{Q}\mathbf{S^T} = \mathbf{C}$. After that, the final transformed

correlated matrix $\mathbf{X_c} = \mathbf{X}\mathbf{S^T}$ can be obtained, which has the desired correlation matrix $\mathbf{C}$. It is noteworthy that the values in

each column of the input N×6 matrix $\mathbf{X}$ can be rearranged to have the same rank-order as the target matrix.

The correlated genesis samples for 100 years for Hong Kong are generated by Monte Carlo simulations coupled with parameter

correlation analysis, as shown in Fig. 13. As can been seen, the observed JMA data points are scattered around the simulated

results. And the correlation coefficients of the simulated variables ($\rho_{sim}$) are almost identical to those of the original

observations ($\rho_{obs}$). It is worth mentioning that the historical data for $\alpha_0$, $\Delta P_0$, $\theta_0$, $V_{T0}$ are more than those for $R_{max,s0}$ and

$B_{s0}$ since the wind speed information is only available from 1977 and the wind data estimations are usually not provided during

the first and last several time steps of a TC track due to its weak intensity. As a result, the scatter plots for historical observations

in Fig. 13 associated with $R_{max,s0}$ and $B_{s0}$ contain fewer data than others. Correspondingly, the correlation coefficients

associated with these two parameters would also be derived from fewer data.

**4.2 Design wind speed prediction**

After generating the virtual tracks as well as the wind field parameters, the TC wind speed at the site of interest can be readily

solved using the wind speed field model. Then, our final objective is to investigate the design wind speeds with various return

intervals or TC wind hazard curves for the site of interest. 10,000-year simulations would be conducted for each site to achieve

adequate TC samples. The underlying terrain exposure is assumed to be consistent with the standard condition specified by

Load Code for the Design of Building Structures (GB-50009 2012), i.e. flat open and low-density residential area of terrain

category B with equivalent roughness length $z_0$ = 0.05 m. These simulated tracks can also be employed to estimate the wind

speed with respect to other underlying exposures by simply using a desired input of $z_0$. And all simulated tracks can be

interpolated into 15 min so as to capture every potential maximum wind speed.

By assuming that number of TCs occurring in a given season is independent of any other season such that the occurrence

probability $P_T(n)$ of $n$ TCs over the time period $T$ can be assumed to follow the Poisson distribution. Then, the probability

that the extreme wind speed $v_i$ is larger than a certain wind speed $V$ within a time period $T$ can be determined as

$$P_T(v_i > V) = 1 - \sum_{n=0}^{\infty} P(v_i \le V|n)P_T(n) = 1 - exp\left(-\frac{N}{Y}T\right),\qquad(11)$$





in which $P(v_i \leq V|n)$ is the probability that the peak wind speed $v_i$ of a given TC is less than or equal to $V$, $N$ is the total

number of TCs that each of them has a peak wind $v_i$ larger than $V$, and $Y$ is total simulation years. Defining $T = 1$ year, the

annual probability of exceeding a given wind speed $V$ is

$$P_{T=1yr}(v_i > V) = 1 - exp[-\lambda P(v_i > V)] = 1 - exp\left(-\frac{N}{Y}\right), \tag{12}$$

in which $\lambda$ is the annual storm occurrence rate within the region of interest. The mean recurrence interval (MRI) or return

period (RP) of a given wind speed $V$ at a specific site can be estimated using the inverse of the result of Eq. (12) with the form

$$RP(v_i > V) = \frac{1}{\lambda P(v_i > V)} = \frac{Y}{N}, \tag{13}$$

Fig. 14 illustrates the empirical distribution of annual maximum TC mean wind speeds (10-min duration at 10-m height) curve

as well as the return period curve of design mean wind speed in Hong Kong. Although the lognormal distribution is adopted

for $\Delta P_0$ in this study, a similar distribution trend of annual maximum TC mean wind speed can be observed in this study and

Li and Hong (2015) when $\Delta P$ is modelled by a Weibull distribution (Fig. 14a). A Weibull distribution was also preferred to

the lognormal distribution in their study. However, the lognormal distribution is the preferred distribution in this study. This

is mainly attributed to the use of different historical track datasets and sub-region size. Li and Hong (2015) adopted the best

track dataset from the China Meteorological Agency and a radius of 250 km for the sub-region circle. Thus, modelling the

historical data with preferable probabilistic distributions is essentially important before the estimation of TC design wind speed

can be regarded as a site-specific issue.

Moreover, Fig. 14b compares the predicted design mean wind speeds with the recommended values in Wind-resistant Design

Specification for Highway Bridges (JTG/T D60-01-204, code hereafter) for different return periods. It can be noted that the

code's values are larger than those obtained in this study and the difference seems to decrease with increase in return period.

This is because the values recommended in the code are developed by statistical approaches based on both TC and non-TC

observations over 30~40 years. Some strong non-TC winds captured by meteorological stations could dominate the design

values for short return periods while strong TC winds would control the higher design wind speed corresponding to longer

return periods.

As mentioned in the explanatory materials to the Hong Kong Code (2004), the 50-year-MRI hourly mean wind speed of

46.9m/s at 90 m above mean sea level with the underlying exposure of open sea was selected as the reference. In this case, the

10-m wind speed is estimated as 36.83 m/s using the power wind profile with the suggested exponent of 0.11 (0.12 for terrain

exposure A in Chinese code, 1/9 for terrain exposure D in ASCE 7-16). The estimated 10-min mean wind speed is roughly

39.04m/s if the conversion factor is 1.06 from 1 hour to 10 min. However, in order to be consistent with the reference exposure

in this study ($z_0 = 0.05$), the gradient wind speed can be determined as 56.64 m/s at 500 m and is assumed to be the same as

other exposures. Then, the 10-min wind speed at height 10 m associated with open flat terrain can be calculated as 33.39 m/s

if the power exponent is 0.15 (0.16 for terrain exposure B in Chinese code, 1/6.5 for terrain exposure C in ASCE 7-16) and





the same gradient height is employed. This value is about 2 m/s smaller than the result of this study (35.16 m/s). Similar results can be found from Kwok (2012), who summarized that the over-sea wind speed at a height of 500 m with an MRI of 50 years was within the range of 54 m/s~57 m/s based on the historical TC records and he recommended a slightly higher value of 59.5 m/s for design purpose. The corresponding 10-min mean wind speed associated with z0 = 0.05 is estimated as 35.07 m/s by

following the same algorithm, which compares favourably to the result in the present study. Accordingly, the predicted design wind speed in Hong Kong in this study has an expected level of confidence for engineering applications.

### 4.3 TC wind hazards at selected coastal cities in China

For comparison with other studies (Xiao et al., 2011; Li and Hong, 2015), nine other coastal cities (Fig. 15), i.e. Shanghai, Ningbo, Wenzhou, Fuzhou, Xiamen, Guangzhou, Shenzhen, Zhanjiang, and Haikou were selected for Monte Carlo simulations

following the aforementioned algorithm. Because the Burr distribution fails to fit the empirical $B_{s0}$ in Shanghai, Ningbo and Wenzhou, the general extreme value (GEV) distribution was employed to model $B_{s0}$ of these three cities. GEV distribution is a commonly used distribution developed from extreme value theory to combine the Gumbel, Fréchet and Weibull function families, also known as types I, II and III extreme value distributions. Its PDF can be expressed as

$$f(x; \mu, \sigma, \gamma) = \frac{1}{\sigma} exp\left[-\left(1 + \gamma \cdot \frac{x-\mu}{\sigma}\right)^{-\frac{1}{\gamma}}\right]\left(1 + \gamma \cdot \frac{x-\mu}{\sigma}\right)^{-1-\frac{1}{\gamma}}, \gamma \neq 0 , \qquad (14)$$

$$f(x; \mu, \sigma, 0) = \frac{1}{\sigma} exp\left[-exp\left(-\frac{x-\mu}{\sigma}\right) - \frac{x-\mu}{\sigma}\right], \gamma = 0 , \qquad (15)$$

in which γ, σ and μ are called shape, scale and location parameters, respectively, and 1+γ(x-μ)/σ > 0. Correspondingly, for γ = 0, γ > 0 and γ < 0 conditions, GEV distributions can be reduced to types I, II and III extreme value distributions. As shown in Table 2~3, coefficients of each distribution for various input parameters in another nine coastal cities of China were estimated using a maximum likelihood method based on historical observation around the site of interest within a radius of

500 km. The annual storm rate was observed to gradually increase from north to south. The fitted coefficients of recursive models of $V_T$, $R_{max,s}$ and $B_s$ as well as the decay model coefficients are also listed in Table 3. Correspondingly, the empirical and fitted preferred CDFs for each parameter in nine cities are illustrated in Fig. 16 together with the K-S test statistics. It can be seen that the distribution models successfully matched the empirical historical samples.

Like Hong Kong, the 10-min mean design wind speeds at height 10 m above the ground with a surface roughness of 0.05 m

with respect to various return periods were developed based on 10,000-year Monte Carlo simulations. Table 4 lists the simulation results for TC design wind speed at selected cities with an MRI of 100 years and compared them with two Chinese codes (JTG/T D60-01-2004; GB 50009-2012) as well as other pioneering studies. The design wind speeds in the two codes are consistent with each other, except for a 2.5 m/s difference in Shanghai. It can be seen that the predicted wind speeds in this study are close to the code-recommended values, except for Ningbo, Wenzhou, Zhanjiang and Haikou. The estimated values

for Ningbo and Wenzhou are more than 4 m/s higher than those in the codes while those for Zhanjiang and Haikou are more





than about 4 m/s smaller. A similar trend can also be observed from the differences between Li and Hong (2016), Chen and Duan (2017) and the codes. This is mainly attributed to the limitations of the statistically short-term data-based method used in the code development. As mentioned before, the design wind speeds in the Chinese codes are developed from short-term observations utilizing both TC and non-TC winds (30~40 years). However, the series of largest annual wind speeds are, in

most cases, not well-behaved (Simiu and Scanlan, 1996) when used for modelling the probabilistic behaviour of the extreme winds since most of the largest annual winds are remarkably smaller than the extreme winds associated with TCs. That is, the contribution of each group of data used for characterizing the probabilistic behaviour of the largest annual winds is uneven, resulting in some unrealistically high or low predictions (Simiu and Scanlan, 1996). Although some alternative approaches can be adopted to better consider TC winds, such as the use of maximum average monthly speed or mixed distributions of TC

and non-TC winds, to the authors' knowledge, no published literature clearly discusses the development of design wind speed in the Chinese codes. Furthermore, correction of averaging time, height, station migration and surrounding roughness to make the wind speed records meteorologically homogeneous would introduce some unpredictable errors.

Moreover, as shown in Fig. 17, violent typhoons ($P_{cs} \leq 935\ hPa$ or $V_{\mathrm{max,s}} \geq 54$ m/s(105 knots) ) as well as strong typhoons ($P_{cs} \leq 960$ hPa or $V_{\mathrm{max,s}} \geq 44$ m/s(85 knots)), that affect Zhanjiang (close to Haikou), Hong Kong (close to Shenzhen),

Wenzhou and Ningbo within 500 km are extracted from the 65-year JMA dataset. It turns out that only two TCs (200814 Hagupit and 201409 Rammasun) around Zhanjiang (or Haikou) and six TCs (195408 Ida, 197909 Hope, 200814 Hagupit, 201013 Megi, 201319 Usagi and 1409 Rammasun) around Hong Kong (or Shenzhen) reached the violent level. Comparatively, 25 and 13 violent typhoons were observed around Wenzhou and Ningbo, respectively. Moreover, 40 and 52 strong typhoons affected Zhanjiang and Hong Kong, respectively, while Wenzhou and Ningbo suffered 89 and 55 strong typhoons over the

past half a century. This is thanks to the obstacle effects of several high mountains in the Philippines so that the violent typhoons making landfall in Hainan and Guangdong provinces usually need to re-intensify in the South China Sea or directly pass through the Bashi Channel between Taiwan and the Philippines, so not many violent typhoons were observed to affect these two provinces. In addition, the maximum wind of the rotating storm in the northern hemisphere always occurs on its right side with respect to the heading direction due to the Coriolis effect. Thus, westward-heading violent typhoons seldom occur in

Zhanjiang and Haikou before their intensities decay due to the effect of Hainan island. Instead, Hong Kong, Wenzhou or Ningbo have greater chances of being swept by a storm's maximum wind. Accordingly, the prediction results should be reasonable with higher design wind speeds in Wenzhou and Ningbo than in Zhanjiang and Haikou. It is suggested that this trend should be validated in a future study using more TC observation data.

The results in Xiao et al. (2011) are higher than those in other studies or codes. There are three possible reasons for this. The

first is the use of the Holland method (2008) in determining $B$ values. This method was developed from semi-empirical relationships between gradient and surface layer as discussed by Fang et al (2018). Another reason is the use of a 1000-km-radius subregion, which would take into account many extremely violent typhoons over the distant sea before they are used for TC intensity modelling. The third one is the use of a surface roughness of 0.02 m, which is smaller than the code-specified value associated with terrain exposure B of 0.05 m.





The estimated wind speeds in Shanghai, Ningbo and Wenzhou are 2~3 m/s higher than Li and Hong (2016) while Zhanjiang showed about a 7 m/s smaller result. The other five cities show a satisfactory comparison between results of this study and Li and Hong (2016). When they are compared with Chen and Duan (2017), who used an improved full track model, the present estimations in Zhanjiang and Haikou are also about 4 m/s smaller while the other cities show 1~4 m/s higher values. Except for the potential reasons analysed above, it is worth mentioning that Li and Hong (2016) adopted CMA track data with 2-min

duration while Chen and Duan (2017) used a JTWC dataset with 1-min duration. Some errors could be introduced by the time duration gaps for different datasets.

Fig. 18 illustrates design wind speed versus return period plots (hazard curves) based on simulations together with the suggested values in Chinese codes (JTG/T D60-01-2004) for nine coastal cities. It can be seen that the predicted curves for Shanghai, Fuzhou, Xiamen, Guangzhou and Shenzhen show satisfactory agreement with code suggestions. But, consistent

with previous findings, this study shows higher estimations for Ningbo and Wenzhou while it shows smaller estimations for Zhanjiang and Haikou than the code. It is also found that the estimated hazard curves for Ningbo and Wenzhou have a similar trend to the code, but the design wind speeds for Zhanjiang and Haikou increase more gently with return period than the code provisions. This is closely related to the portion of TC wind samples as well as their contributions to the description of the probabilistic distribution of extreme winds in a series of largest observed annual winds as discussed above. The TC winds in

Ningbo and Wenzhou could dominate the probabilistic behaviour of the yearly largest wind speed while Zhanjiang and Haikou have lower portions of TC winds compared to synoptic winds. However, the contributions of strong TC winds will be overused in modelling the hazard curve when they are combined with smaller synoptic winds in the yearly largest wind series. More observations on TC winds and unique descriptions of the probabilistic behaviour of TC winds are necessary to model site-specific TC hazards and validate the long-term hazard predictions in this study.

**5 Conclusions**

The statistical characteristics of TC track as well as wind field parameters within a site-specific circular subregion extracted from the JMA best track dataset were examined before developing TC wind speed hazard curves for 10 coastal cities in China using a height-resolving wind field model and a Monte Carlo technique. Some improvements and new findings are summarized as follows:

(1) Recursive models are applied for both track (translation speed) and wind field ($R_{max,s}$ and $B_s$) parameters, which enable the movement as well as the size and wind field scale of a TC to vary smoothly. $R_{max,s}$ and $B_s$ of the historical dataset are determined from the present height-resolving wind field model coupled with 10-min-duration wind information provided by JMA. Thus, the present study is self-adaptive, and no other statistical models of wind field parameters are adopted, which are commonly cross used in other studies. Meanwhile, the documented $R_{max,s}$ and $B_s$ dataset facilitates the completeness of

correlation studies between various parameters at first time steps before generating statistically correlated parameters using the Cholesky decomposition method.

(2) The probabilistic behaviour of TC track and wind model parameters of the first time steps (genesis parameters) within a 500-km circular subregion of 10 coastal cities are investigated and modelled with some preferable probability distribution models. Then the coefficients of the decay model as well as the recursive models for translation speed, $R_{max,s}$ and $B_s$ in these 10 cities are also fitted.

(3) The TC design wind speed versus return period plots (hazard curve) are developed from 10,000-year Monte Carlo simulations and compared with code suggestions as well as other studies. It is found that the predicted wind speeds in northern cities (Ningbo and Wenzhou) are higher than code suggestions while those of southern cities (Zhanjiang and Haikou) are smaller. The other six cities show satisfactory agreement with code provisions. Some potential reasons for this are discussed to emphasize the importance of independently developing hazard curves of TC and synoptic winds.

*Data availability*. All data that support the findings of this study are available from the corresponding author by request.

*Author contributions*. GF performed the simulations and data analyses. LZ developed the methodology. GF and LZ wrote the original draft. SC and LZ reviewed and edited the manuscript. YG guided intellectual direction of the research.

*Competing interests*. The authors declare no competing interests.

*Acknowledgements*. The authors gratefully acknowledge the support of the National Key Research and Development Program of China (2018YFC0809600, 2018YFC0809604) and the National Natural Science Foundation of China (51678451, 51778495).

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


**Table 1: Distribution models and coefficients for TC track genesis parameters**

| Parameter | Model | Probability density function (PDF) | Coefficient (Hong Kong) |
|---|---|---|---|
| $\lambda_a$ | Poisson | $f(x;\lambda) = \dfrac{\lambda^x}{x!}e^{-\lambda}, \quad x = 0,1,2,\cdots,\infty$ | $\lambda = 6.339$ |
| $\alpha_0$ | Weibull | $f(x;k,\gamma) = \dfrac{k}{\gamma}\left(\dfrac{x}{\gamma}\right)^{k-1}e^{-(x/\gamma)^k}, \quad x \geq 0$ | $k = 3.134; \gamma = 156.991$ |
| $\Delta P_0$ | Lognormal | $f(x;\mu,\sigma) = \dfrac{1}{x\sigma\sqrt{2\pi}}exp\left\{\dfrac{-(lnx-\mu)^2}{2\sigma^2}\right\}, \quad x > 0$ | $\mu = 3.062; \sigma = 0.576$ |
| $\theta_{T0}$ | Bimodal normal | $f(x;p,\mu_1,\sigma_1,\mu_2,\sigma_2)$ $= p\dfrac{1}{\sigma_1\sqrt{2\pi}}exp\left\{\dfrac{-(x-\mu_1)^2}{2\sigma_1^2}\right\}$ $+ (1-p)\dfrac{1}{\sigma_2\sqrt{2\pi}}exp\left\{\dfrac{-(x-\mu_2)^2}{2\sigma_2^2}\right\}$ | $p = 0.475; \mu_1 = -73.282; \sigma_1 = 25.607; \mu_2 = 0.002; \sigma_2 = 68.030;$ |
| $V_{T0}$ | Burr type XII | $f(x;\alpha,c,k) = \dfrac{\frac{kc}{\alpha}\left(\frac{x}{\alpha}\right)^{c-1}}{\left(1+\left(\frac{x}{a}\right)^c\right)^{k+1}},$ $x > 0, \alpha > 0, c > 0, k > 0$ | $\alpha = 16.151, c = 2.540, k = 15.028$ |

Note: $x$ denotes the argument or the input of the function.

**Table 2: Coefficients of PDFs for TC track genesis parameters**

| City | Lat (°N) | Lon (°E) | $\lambda_a$ | $\alpha_0$ | | $\Delta P_0$ | |
|---|---|---|---|---|---|---|---|
| | | | $\lambda$ | $k$ | $\gamma$ | $\mu$ | $\sigma$ |
| Shanghai | 31.233 | 121.483 | 3.139 | 4.160 | 182.519 | 3.119 | 0.668 |
| Ningbo | 29.867 | 121.517 | 3.662 | 3.901 | 180.383 | 3.204 | 0.691 |
| Wenzhou | 28.017 | 120.650 | 4.600 | 3.697 | 176.511 | 3.236 | 0.703 |
| Fuzhou | 26.083 | 119.300 | 4.923 | 3.121 | 172.821 | 3.201 | 0.634 |
| Xiamen | 24.483 | 118.100 | 5.615 | 3.301 | 170.379 | 3.177 | 0.650 |
| Guangzhou | 23.000 | 113.217 | 5.677 | 3.336 | 155.768 | 3.034 | 0.566 |
| Shenzhen | 22.550 | 114.117 | 6.154 | 3.220 | 157.946 | 3.062 | 0.581 |
| Hong Kong | 22.300 | 114.167 | 6.339 | 3.134 | 156.991 | 3.062 | 0.576 |
| Zhanjiang | 21.271 | 110.361 | 5.569 | 3.316 | 138.980 | 3.040 | 0.554 |
| Haikou | 20.367 | 110.333 | 5.862 | 3.291 | 132.367 | 3.049 | 0.563 |






**Table 2 (Cont.): Coefficients of PDFs for TC track genesis parameters**

| City | $\theta_{T0}$ | | | | | $V_{T0}$ | | |
|---|---|---|---|---|---|---|---|---|
| | $p$ | $\mu_1$ | $\sigma_1$ | $\mu_2$ | $\sigma_2$ | $\alpha$ | $c$ | $k$ |
| Shanghai | 0.201 | -61.625 | 32.169 | 21.807 | 38.422 | 7.407 | 3.321 | 1.576 |
| Ningbo | 0.193 | -68.056 | 36.079 | 11.396 | 44.951 | 6.879 | 3.738 | 1.531 |
| Wenzhou | 0.107 | -68.363 | 19.573 | -7.533 | 57.165 | 7.405 | 3.605 | 1.813 |
| Fuzhou | 0.190 | -67.363 | 23.536 | -8.797 | 23.536 | 7.988 | 3.284 | 2.788 |
| Xiamen | 0.267 | -70.547 | 25.815 | -4.259 | 59.630 | 7.774 | 3.167 | 2.969 |
| Guangzhou | 0.506 | -72.845 | 28.000 | 0.002 | 66.048 | 9.651 | 2.765 | 4.777 |
| Shenzhen | 0.460 | -73.308 | 25.226 | -3.249 | 67.401 | 31.878 | 2.449 | 67.578 |
| Hong Kong | 0.475 | -73.282 | 25.607 | 0.002 | 68.030 | 16.151 | 2.540 | 15.028 |
| Zhanjiang | 0.614 | -74.773 | 25.304 | -3.412 | 70.905 | 15.400 | 2.734 | 14.735 |
| Haikou | 0.620 | -75.013 | 24.847 | -5.740 | s73.308 | 11.820 | 2.799 | 7.926 |

**Table 3: Coefficients of PDFs and recursive models for wind field parameters**

| City | $R_{max,s0}$ | | $B_{s0}$ | | | $V_T$ | | | |
|---|---|---|---|---|---|---|---|---|---|
| | $\mu$ | $\sigma$ | $\alpha(\mu)$ | $c(\sigma)$ | $k(k)$ | $v_1$ | $v_2$ | $v_3$ | $v_4$ |
| Shanghai | 5.062 | 0.665 | 1.850 | 0.501 | -0.542 | 0.325 | 0.702 | 0.129 | 1.283e-3 |
| Ningbo | 5.064 | 0.640 | 1.839 | 0.479 | -0.523 | 0.319 | 0.689 | 0.147 | 1.273e-3 |
| Wenzhou | 4.905 | 0.628 | 1.705 | 0.440 | -0.368 | 0.273 | 0.644 | 0.209 | 9.689e-4 |
| Fuzhou | 4.831 | 0.567 | 2.055 | 6.439 | 2.247 | 0.344 | 0.602 | 0.201 | 8.444e-4 |
| Xiamen | 4.805 | 0.591 | 1.850 | 7.198 | 1.412 | 0.358 | 0.590 | 0.196 | 7.724e-4 |
| Guangzhou | 4.802 | 0.598 | 1.779 | 6.895 | 1.321 | 0.305 | 0.612 | 0.179 | 1.304e-4 |
| Shenzhen | 4.817 | 0.631 | 2.610 | 5.154 | 5.936 | 0.303 | 0.635 | 0.154 | 1.129e-4 |
| Hong Kong | 4.822 | 0.571 | 1.974 | 6.362 | 2.001 | 0.309 | 0.634 | 0.150 | 1.094e-4 |
| Zhanjiang | 4.830 | 0.571 | 1.545 | 8.526 | 0.765 | 0.276 | 0.610 | 0.181 | -3.284e-4 |
| Haikou | 4.813 | 0.575 | 1.529 | 9.024 | 0.713 | 0.282 | 0.610 | 0.179 | -3.499e-4 |




**Table 3 (Cont.): Coefficients of PDFs and recursive models for wind field parameters**

| City | $R_{max,s}$ | | | | $B_s$ | | | | | $a$ | |
|------|-------|-------|--------|--------|--------|-------|--------|-------|---------|-------|---------|
| | $r_1$ | $r_2$ | $r_3$ | $r_4$ | $b_1$ | $b_2$ | $b_3$ | $b_4$ | $b_5$ | $a_1$ | $a_2$ |
| Shanghai | 0.544 | 0.866 | 0.037 | -1.172e-3 | -1.104 | 0.327 | 0.041 | 0.449 | -1.172e-3 | 0.020 | 5.026e-4 |
| Ningbo | 0.510 | 0.856 | 0.056 | -1.359e-3 | -0.870 | 0.369 | 0.040 | 0.390 | -1.359e-3 | 0.014 | 6.083e-4 |
| Wenzhou | 0.668 | 0.871 | 0.018 | -1.886e-3 | -0.918 | 0.420 | -0.027 | 0.403 | 1.538e-3 | 0.024 | 4.430e-4 |
| Fuzhou | 0.637 | 0.899 | -2.888e-3 | -2.013e-3 | -0.899 | 0.394 | -0.020 | 0.404 | 1.770e-3 | 0.024 | 4.242e-4 |
| Xiamen | 0.657 | 0.910 | -0.023 | -1.592e-3 | -0.804 | 0.469 | -0.057 | 0.374 | 1.179e-3 | 0.024 | 4.787e-4 |
| Guangzhou | 0.727 | 0.824 | 0.032 | -1,646e-3 | -0.626 | 0.537 | -0.022 | 0.298 | 4.951e-4 | 0.022 | 5.801e-4 |
| Shenzhen | 0.703 | 0.813 | 0.039 | -3.815e-4 | -0.603 | 0.574 | 0.001 | 0.269 | 6.182e-4 | 0.026 | 5.201e-4 |
| Hong Kong | 0.704 | 0.834 | 0.028 | -1.630e-3 | -0.665 | 0.543 | -0.011 | 0.295 | 1.300e-3 | 0.022 | 5.654e-4 |
| Zhanjiang | 0.703 | 0.813 | 0.039 | -3.815e-4 | -0.603 | 0.574 | 0.001 | 0.269 | 6.182e-4 | 0.026 | 5.201e-4 |
| Haikou | 0.680 | 0.803 | 0.054 | -4.531e-4 | -0.642 | 0.558 | 0.011 | 0.275 | 1.167e-3 | 0.028 | 5.184e-4 |

**Table 4: Comparison of TC design wind speed at selected cities (MRI = 100 year; T = 10 min; z = 10 m, z0 = 0.05m, m/s)**

| City | JTG/T D60-01-2004 | GB 5009-2012 | Xiao et al. (2011) | Li and Hong (2016) | | Chen and Duan (2017) | This study |
|------|-------------------|--------------|--------------------|--------------------|------|----------------------|------------|
| | | | | CSM | FTM | | |
| Shanghai | 33.8 | 31.30 | 48.27 | 32.2 | 31.7 | 31.7 | 34.35 |
| Ningbo | 31.3 | 31.30 | 44.93 | 33.3 | 33.0 | 34.5 | 35.33 |
| Wenzhou | 33.8 | 33.81 | 48.75 | 36.1 | 36.5 | 34.9 | 39.21 |
| Fuzhou | 37.4 | 37.25 | 48.47 | 37.8 | 35.1 | 33.6 | 37.41 |
| Xiamen | 39.7 | 39.38 | 46.70 | 39.1 | 38.9 | 37.7 | 39.18 |
| Guangzhou | 31.3 | 31.30 | 41.57 | 30.5 | 31.4 | − | 30.87 |
| Shenzhen | 38.4 | 38.33 | 43.79 | 36.4 | 36.8 | 36.4 | 37.34 |
| Hong Kong | 39.5 | 39.38 | 45.03 | 37.6 | 37.7 | − | 38.17 |
| Zhanjiang | 39.4 | 39.38 | 42.86 | 40.9 | 37.4 | 37.5 | 33.92 |
| Haikou | 38.4 | 38.33 | 42.94 | − | − | 38.5 | 34.52 |




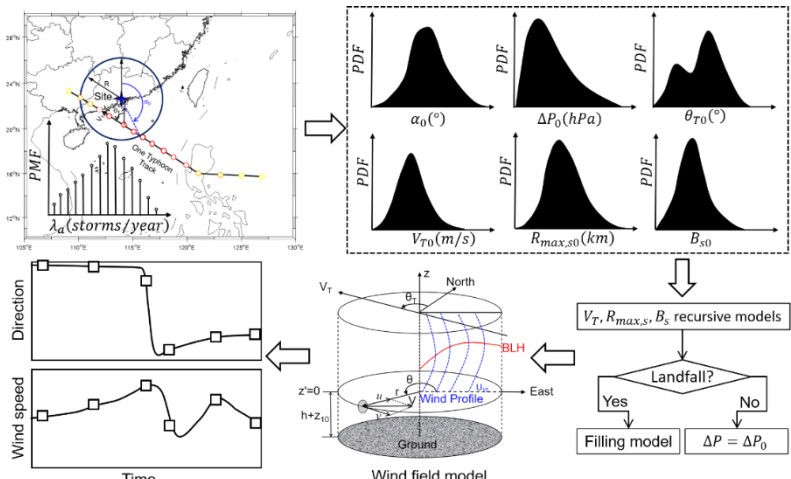


**Figure 1: Overview of circular sub-region method used in this study**

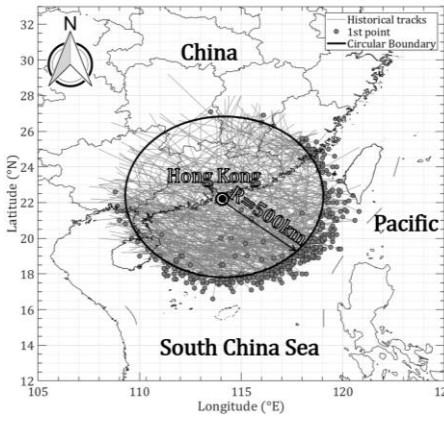

**Figure 2: Track segments within a circular region entered on Hong Kong with a radius of 500 km**

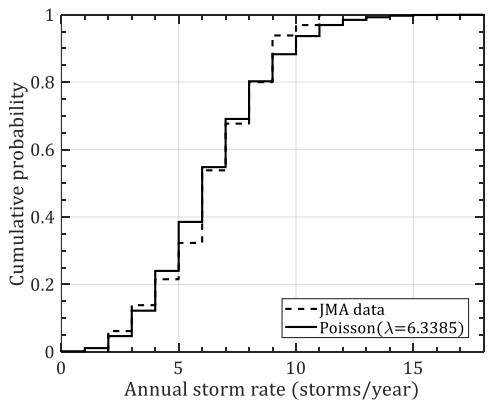

**Figure 3: CDF of annual storm rate**




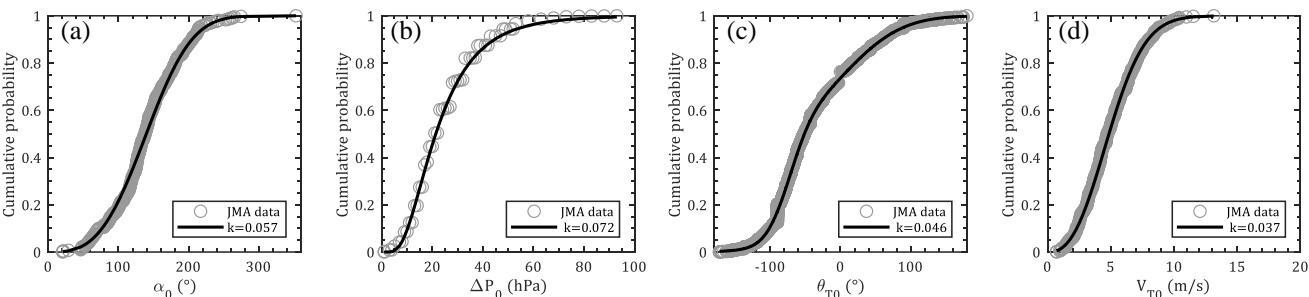

**Figure 4: CDFs of track genesis parameters: (a) $\alpha_0$; (b) $\Delta P_0$; (c) $\theta_{T0}$; (d) $V_{T0}$**

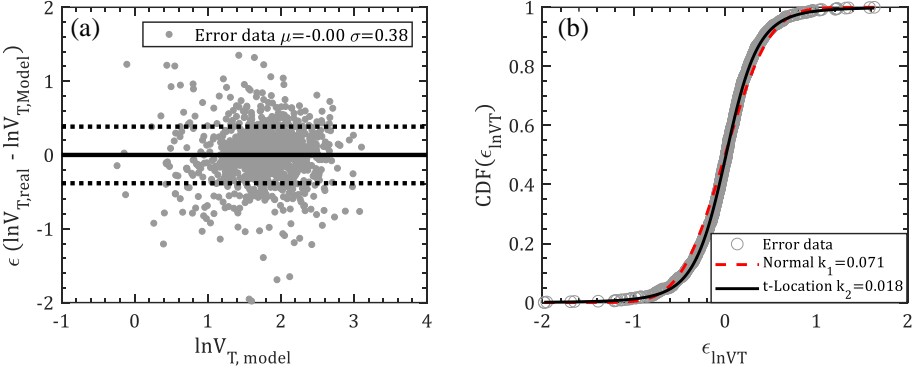

**Figure 5: Logarithmic modelling errors for translation speed: (a) scatter plot; (b) CDF**

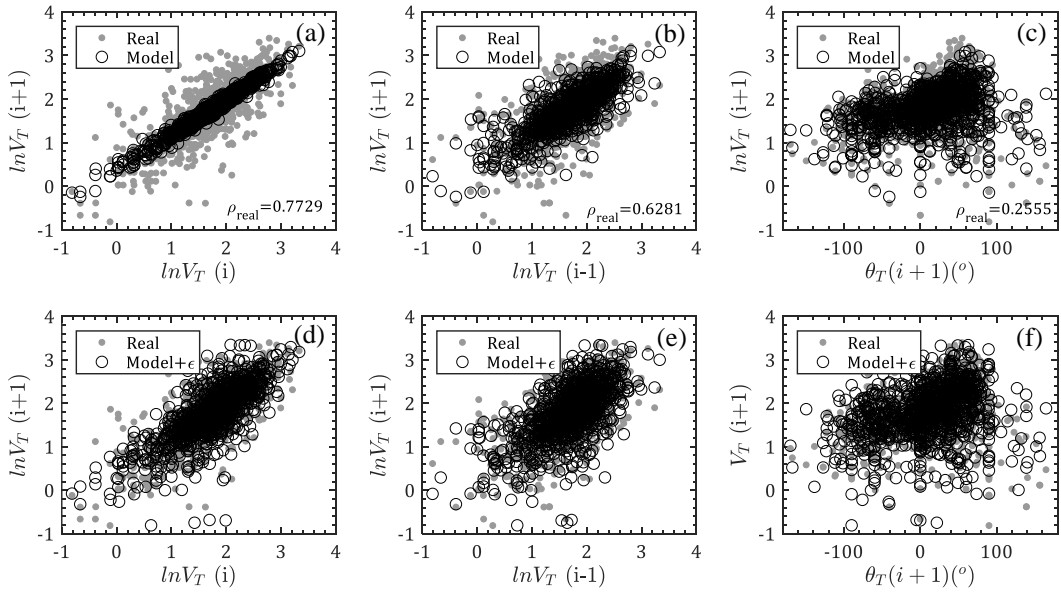

**Figure 6: Comparison of translation speed between model and real observations: (a~c) relations between $lnV_T(i)$, $lnV_T(i-1)$, $\theta(i+1)$ and $lnV_T(i+1)$ without errors; (d~f) relations between $lnV_T(i)$, $lnV_T(i-1)$, $\theta(i+1)$ and $lnV_T(i+1)$ with errors; ($\rho_{real}$ is the correlation coefficient for real observation data)**





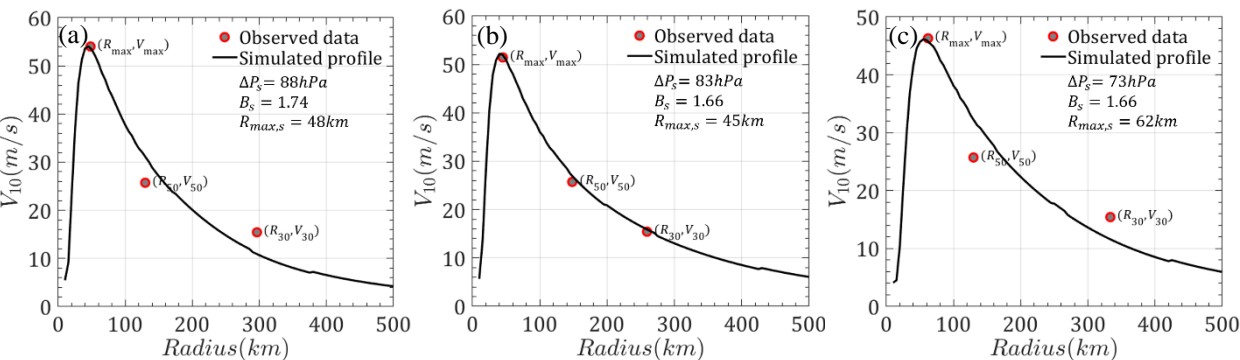

**Figure 7: Radial wind speed profiles (a) Saomai (2006-08-09, 15:00UTC); (b) Parma (2009-10-01, 06:00UTC); (c) Rammasun (2014-07-18, 12:00UTC)**

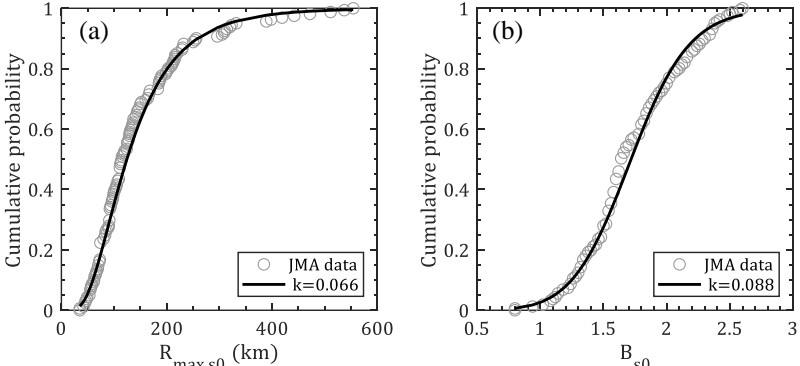

**Figure 8: CDFs of wind field parameters at first step: (a) $R_{max,s0}$; (b) $B_{s0}$**

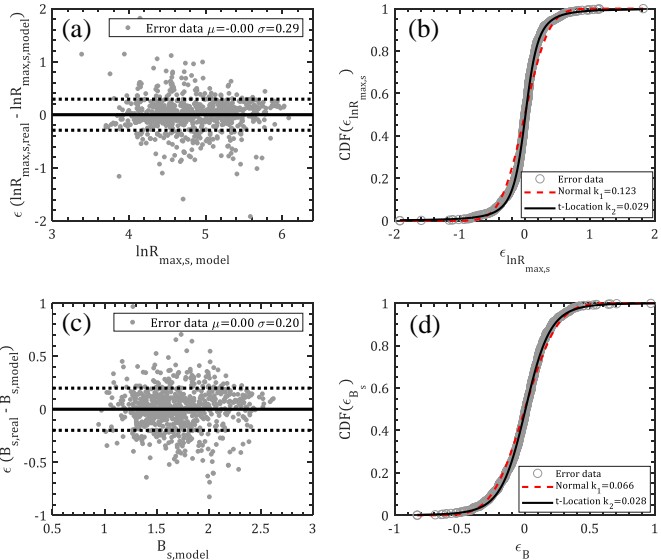

**Figure 9: Model errors for $lnR_{max,s}$ and $B_s$: (a) scatter plot $\left(\varepsilon_{lnR_{max,s}}\right)$; (b) CDF $\left(\varepsilon_{lnR_{max,s}}\right)$; (c) scatter plot $\left(\varepsilon_{B_s}\right)$; (d) CDF $\left(\varepsilon_{B_s}\right)$;**




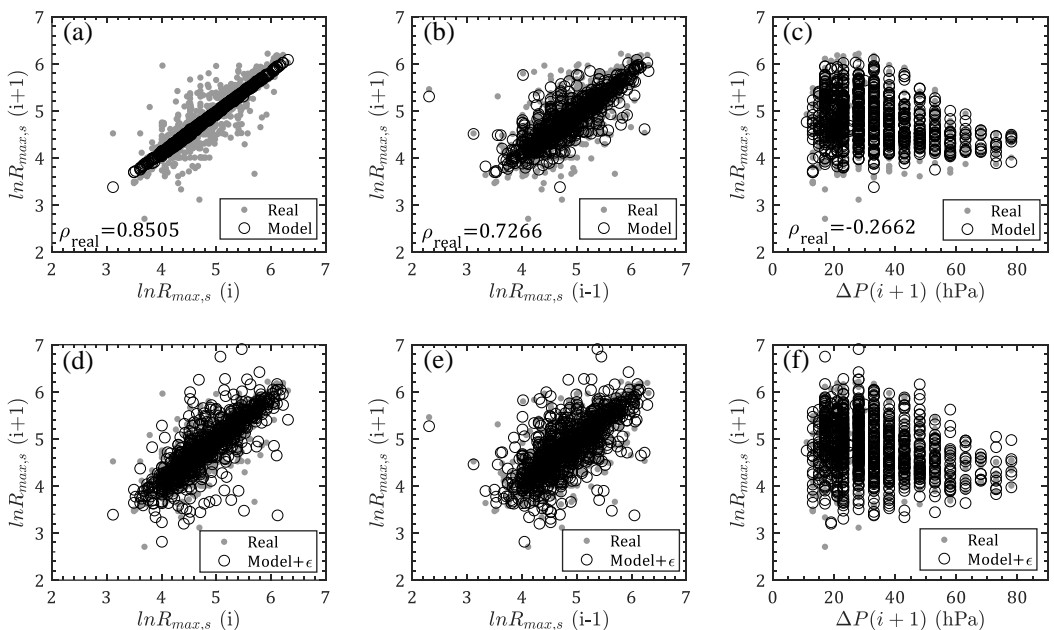

**Figure 10: Comparison of $lnR_{max,s}$ between model and real observations: (a~c) relations between $lnR_{max,s}(i)$, $lnR_{max,s}(i-1)$, $\Delta P(i+1)$ and $lnR_{max,s}(i+1)$ without errors; (d~f) relations between $lnR_{max,s}(i)$, $lnR_{max,s}(i-1)$, $\Delta P(i+1)$ and $lnR_{max,s}(i+1)$ with errors ($\rho_{real}$ is the correlation coefficient for real observation data)**

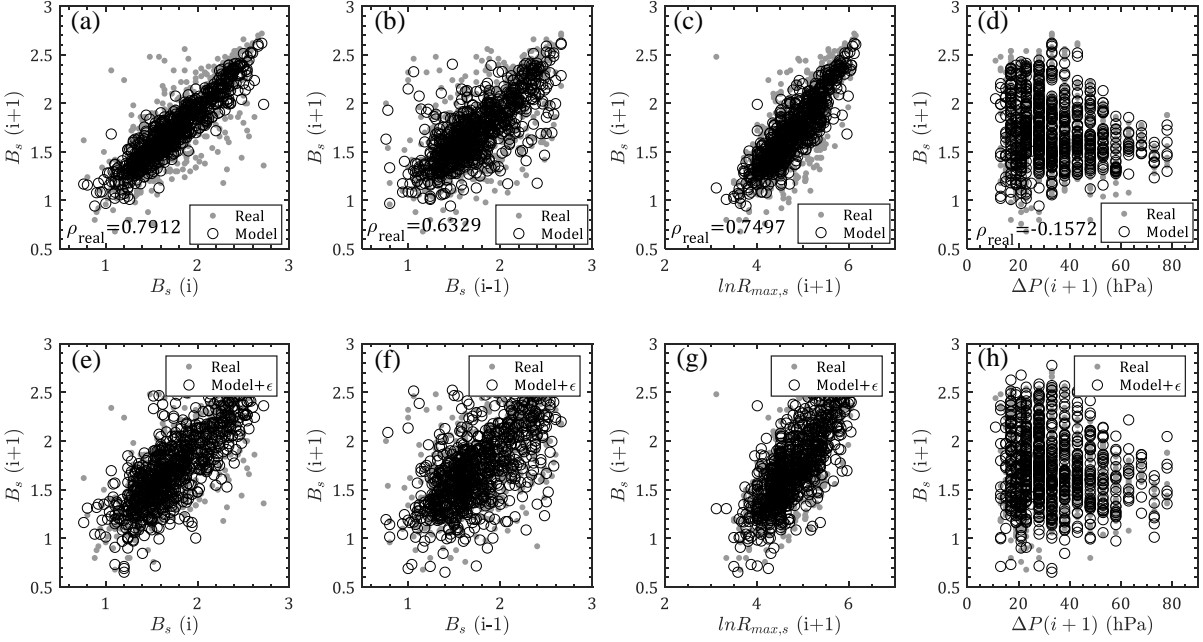


**Figure 11: Comparison of $B_s$ between model and real observations: (a~d) relations between $B_s(i)$, $B_s(i-1)$, $lnR_{max,s}(i+1)$, $\Delta P(i+1)$ and $B_s(i+1)$ without errors; (e~h) relations between $B_s(i)$, $B_s(i-1)$, $lnR_{max,s}(i+1)$, $\Delta P(i+1)$ and $B_s(i+1)$ with errors ($\rho_{real}$ is the correlation coefficient for real observation data)**





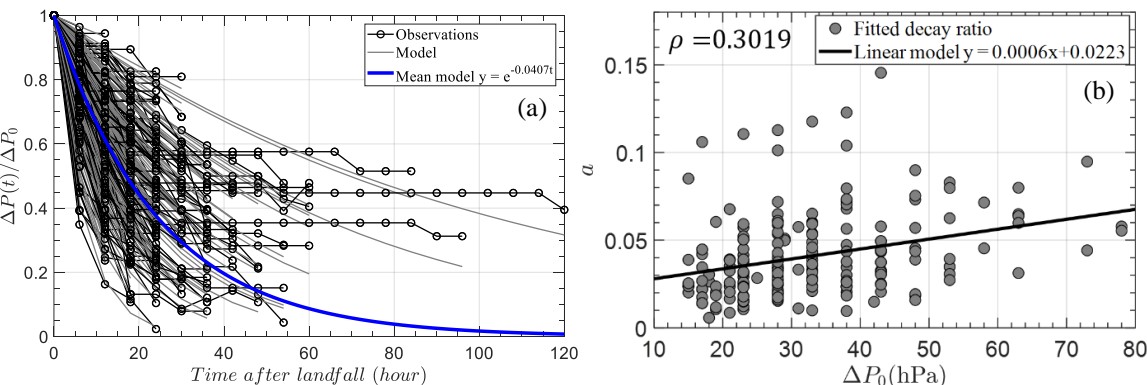

**Figure 12: Decay model in circular sub-region around Hong Kong:(a) Curve fitting of decay model; (b) Decay rate versus $\Delta P_0$**

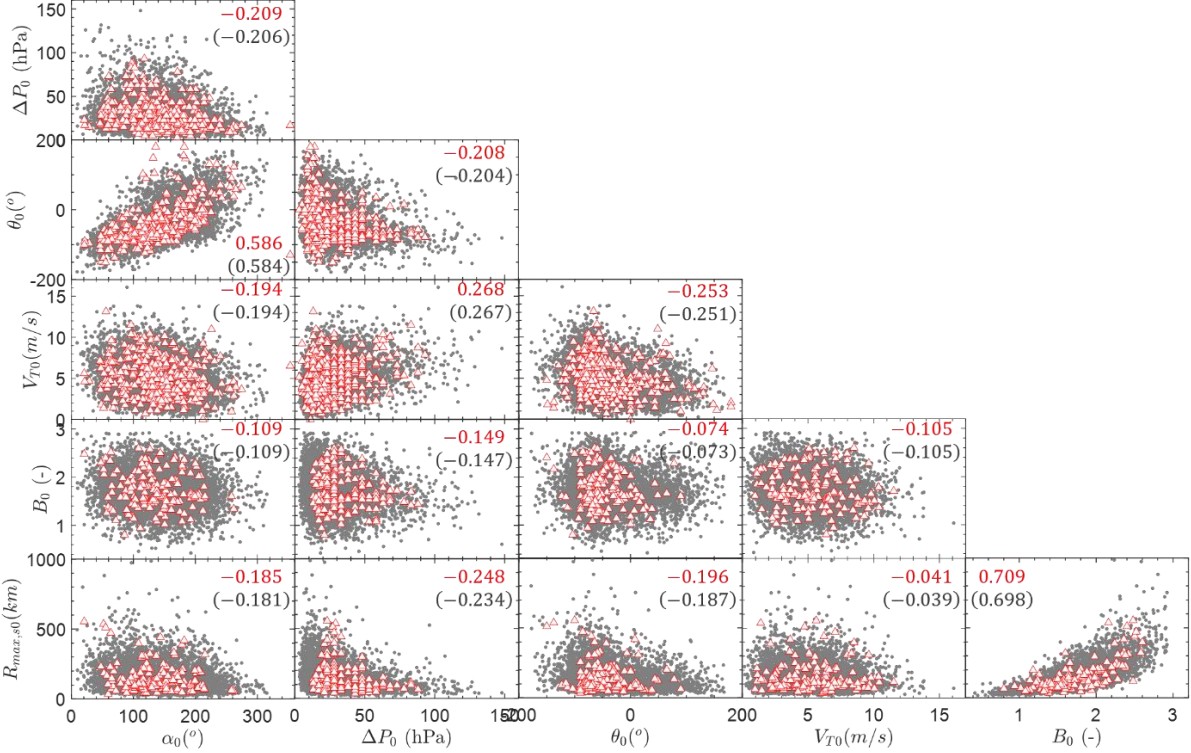

**Figure 13: Simulated and observed genesis parameters (Red triangles: observations; Grey dots: simulations; Upper numbers: $\rho_{sim}$; Lower numbers in parenthesis: $\rho_{obs}$;)**


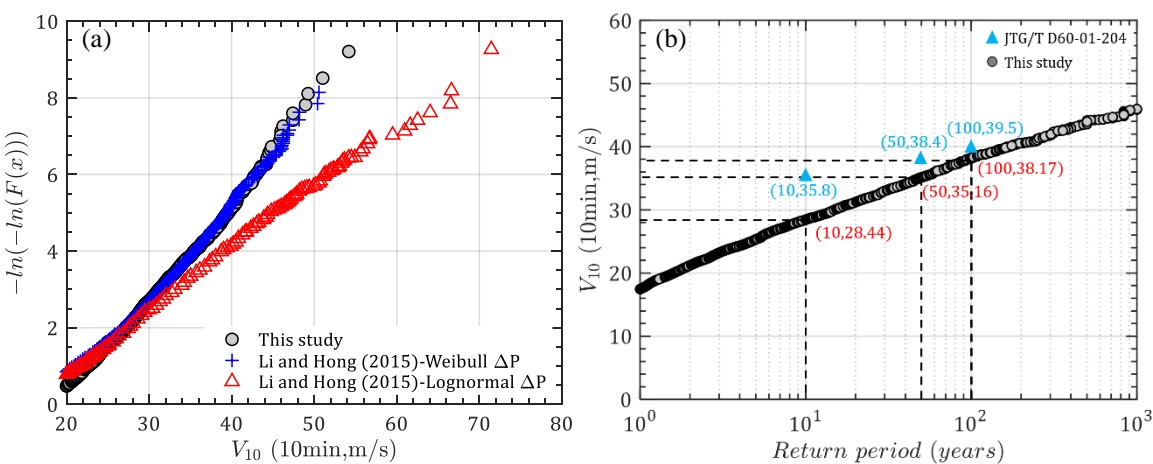

**Figure 14: Design mean wind speed in Hong Kong: (a) Empirical distribution; (b) Mean wind speed versus return periods**

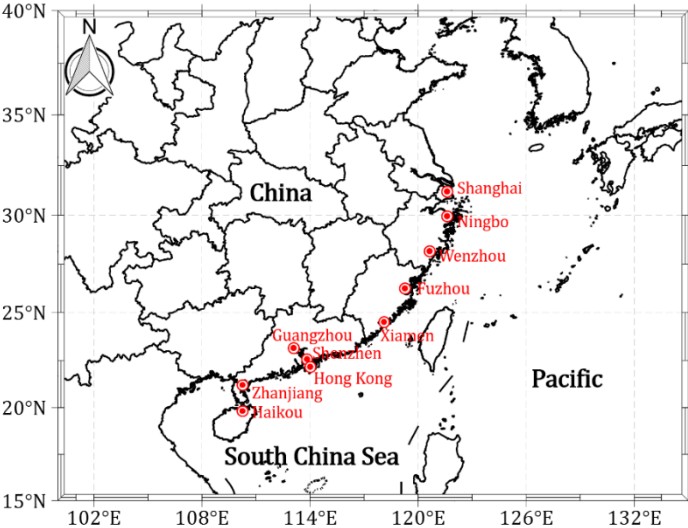

**Figure 15: Locations of 10 selected coastal cities in China**



Natural Hazards
and Earth System




**Figure 16: Empirical and preferable cumulative probability distributions for $\alpha_0, \Delta P_0, \theta_0, V_{T0}, R_{max,s0}$ and $B_{s0}$**






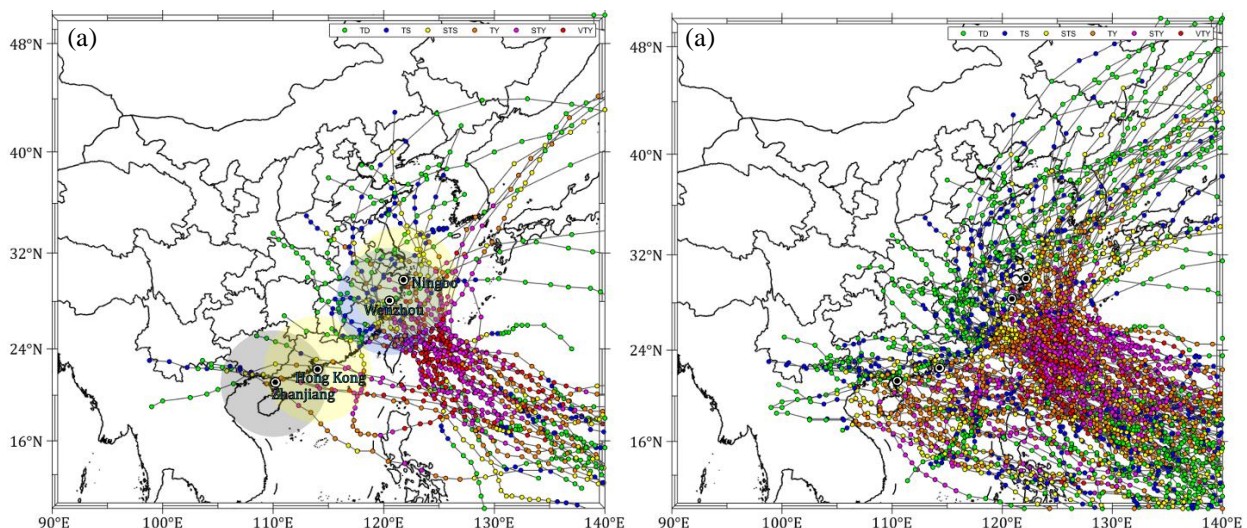

**Figure 17:** Strong typhoon tracks affect Ningbo, Wenzhou, Hong Kong and Zhanjiang: (a) Violent typhoons ($P_c < 935\ hPa$ or $V_{max} > 57\ m/s$); (b) Strong typhoons ($P_c < 960\ hPa$ or $V_{max} > 43\ m/s$)



**Figure 18: Predicted and code-suggested TC design wind speed versus return period of nine coastal cities in China**
