# Peer review of "Estimation of Tropical Cyclone Wind Hazards in Coastal Regions of China"

_Natural Hazards and Earth System Sciences, 2019_

## Referee Comment (RC1) · Anonymous Referee #1 · 17 Feb 2020

The manuscript presents an interesting study on the estimation of tropical cyclone wind hazards. The topic falls in the scope of Natural Hazards and Earth System Sciences (NHESS). Generally, the paper is well written and organized. Some new findings different from suggestions in current specifications are highlighted and discussed. The presented research is of great importance to the wind-resistant design in coastal areas of China. The manuscript can be accepted for publication after minor revisions.

The reviewer has the following concerns for the revisions of the manuscript.

1. The values of the shape parameter of radial pressure profile in Fig. 11. Holland (1980) suggested that it should fall in the range [1.0, 2.5]. Vickery et al. (2000) suggested the range should be [0.5, 2.5]. There are a number of points larger than 2.5 in Fig. 11, which goes against our conventional cognition. Please give some essential

explanations to clarify this point. i) Holland, G. J.: An analytic model of the wind and pressure profiles in hurricanes, Monthly Weather Review, 108, 1212-1218, 1980. ii) Vickery, P. J., Skerlj, P. F., Steckley, A. C., and Twisdale, L. A.: Hurricane Wind Field Model for Use in Hurricane Simulations, Journal of Structural Engineering, 126, 1203-1221, 2000.

2. Fig. 11 can be improved to avoid some data points obscured by legend.

3. Lines 24, 37, 40, 416, 440: characterizing tropical cyclone as 'non-synoptic' is questionable. Tropical cyclone is actually a non-frontal synoptic-scale cyclone as discussed by Vallis et al (2019). Vallis, M. B., Loredo-Souza, A. M., Ferreira, V., Nascimento E. L.: Classification and identification of synoptic and non-synoptic extreme wind events from surface observations in South America, Journal of Wind Engineering and Industrial Aerodynamics, 193, 2019, 103963.

4. Although this paper focuses on the characteristics of the mean components of tropical cyclones, some discussions on the fluctuation components (stationary or non-stationary) are suggested to be supplemented in the introduction part. The following references may do some help. i) Modelling of longitudinal evolutionary power spectral density of typhoon winds considering high-frequency subrange. Journal of Wind Engineering and Industrial Aerodynamics 2019, 193, 103957. ii) Reduced-Hermite bifold-interpolation assisted schemes for the simulation of random wind field. Probabilistic Engineering Mechanics 2018, 53, 126-142.

5. There are some typos in the manuscript, e.g., In line 124, "influnence" should be "influence"; In line 149, "modeling" was used while "modelling" was utilized in line 154. Please use a consistent form.

---

## Short Comment (SC1) · 19 Feb 2020

Comments on Manuscript submitted to NHESS which is entitled "Estimation of Tropical Cyclone Wind Hazards in Coastal Regions of China":

General comments: This article presents a detailed study on the estimation of TC-wind hazards in southeast coast of China. Values of key parameters of TCs, i.e., RMW and Holland-B, are firstly estimated by fitting TC best-track records from JMA via a TC wind field model. These results are then utilized to generate a number of recursive models for corresponding parameters of TC activities and TC wind field. The proposed recursive models are further exploited in conjunction of the TC wind field model to estimate TC extreme winds associated with different return periods at several selected coastal

cities. Finally, results of TC wind hazards obtained from this study are compared with those stipulated in codes or the ones documented by peers. Overall, this work is well written and the analysis process is scrupulous, which makes the findings convincing. It is expected that the findings can provide further insights to better understand the design speeds at coastal areas of China. This reviewer actually has few specific comments for the improvement of this article, but there are still some issues that should be clarified.

Detailed comments: 1. RMW and Holland-B are two key parameters whose values can influence the simulation results of TC wind field severely. Actually, some researchers pointed out that the majority of uncertainty for assessing TC wind hazards should be attributed to the estimation of these parameters. In this regard, great efforts are encouraged to pay to accurate estimation of their values. Basically, there are two kinds of methods which are driven by wind speed records and pressure records, respectively. According to the pioneering work by Holland (1980), RMW and Holland-B are defined under the context of TC pressure field, which potentially indicates that the pressure-data driven method is more straightforward, and possibly more effective. As stated in my general comments, the authors choose the speed-data driven method. Besides the above consideration, there are also several uncertainty sources: (1) even though the authors explain much for choosing TC records from JMA, the basic records herein still belong to the "best-track" data, which means they may differ from the real noticeably. (2) TC wind field possesses asymmetric features, while according to the statements in this study, the best-track information for estimating these two parameters may practically account for symmetric TC wind field. If this is the case, the estimation accuracy could be degraded. (3) The authors use a height-resolving model to depict TC wind field, while the best-track TC information is given at a fixed level. Please detail in the context how to deal with the inconsistency in terms of height level between model and dataset (including what altitude should the best-track data best account for). It is also suggested that the obtained values of RMW and Holland-B be statistically compared with their counterparts in previous

studies. 2. Why do the authors choose a height-resolving TC wind field model rather than others, e.g., a slab model, in this study? To match it with the best-track data which account for a height beyond near ground range? Please clarify. 3. Another comment is about the gradient height. It is assumed in this study that the gradient height is equal to 500 m. However, observational results show that TC depth tends to deepen when TCs get close to coastal areas. Will the inaccuracy of TC depth influence the estimation results? If so, to what an extent? 4. Some minor comments: 1) Line 21: under TC climates climate 2) Lines 225-226: The critical value of K-S test (n = 161) is 0.1059 at a 5% significance level larger than the test statistics...

Please also note the supplement to this comment:
https://www.nat-hazards-earth-syst-sci-discuss.net/nhess-2019-375/nhess-2019-375-SC1-supplement.pdf

---

## Referee Comment (RC2) · Guoqing Huang (Referee) · 8 Apr 2020

This manuscript estimates the tropical cyclone wind hazards in southeastern coastal region of China. Two typhoon wind field parameters, i.e. radius to maximum winds $R_{(max,s)}$ and shape parameter of radial pressure profile $B_s$ are identified using JMA best track dataset coupled with a boundary layer wind field model. TC wind hazard curves in terms of design wind speed versus return periods for major coastal cities of China are developed. The topic of this study is in-line with the journal of "Natural Hazards and Earth System Sciences (NHESS)". Generally, the paper is a well-organized study and worth to be published. The obtained results will be valuable to the researchers and engineers in this field.

This reviewer has minor comments about this study as follows: (1) The major concern is the use of the wind-driven $R_{(max,s)}$ and $B_s$. The results in Figs. 11 and 13 show that $B_s$ and $R_{(max,s)}$ have a positive correlation which is inconsistent with the findings by Vickery et al. (2008). And few values of $B_s$ are higher than 2.5 which fall outside the range of 0.5~2.5 suggested by Vickery et al. (2000). Please explain. Vickery, P. J., Skerlj, P. F., Steckley, A. C., and Twisdale, L. A.: Hurricane Wind Field Model for Use in Hurricane Simulations, Journal of Structural Engineering, 126, 1203-1221, 2000. Vickery, P. J. and Wadhera, D.: Statistical Models of Holland Pressure Profile Parameter and Radius to Maximum Winds of Hurricanes from Flight-Level Pressure and H*Wind Data, Journal of Applied Meteorology and Climatology, 47, 2497-2517, 2008.

(2) The titles of section 2.1 and 2.2 are identical. Please check.

(3) Line 409, "...show satisfactory agreement with...", consider use "...show a satisfactory agreement with..." or "...are in satisfactory agreement with...".

(4) A similar study performed by Wu and Huang (2019) is suggested to be compared and discussed. Wu F., and Huang G.: Refined Empirical Model of Typhoon Wind Field and Its Application in China, Journal of Structural Engineering, 145(11): 04019122, 2019.

---

## Author Comment (AC1) · 12 Apr 2020

Dear Anonymous Referee #1,

We would like to thank you for your constructive comments to the manuscript. We agree with all your comments and we have revised the manuscript accordingly. We are already crafting a revised version of the paper. Please, find below the referees' comments repeated in italics and our responses inserted after each comment. Attached please find our responses.

Regards, Authors

Please also note the supplement to this comment:

[Figure]

https://www.nat-hazards-earth-syst-sci-discuss.net/nhess-2019-375/nhess-2019-375-AC1-supplement.pdf

**Supplement:**

**Responses to *Anonymous Referee #1' Comments**

**Manuscript Number: nhess-2019-375**

**Title of Paper:** Estimation of Tropical Cyclone Wind Hazards in Coastal Regions of China

**Journal:** Natural Hazards and Earth System Sciences (NHESS)

Dear Anonymous Referee #1,

We would like to thank you for your constructive comments to the manuscript. We agree with all your comments and we have revised the manuscript accordingly. We are already crafting a revised version of the paper. Please, find below the referees' comments repeated in italics and our responses inserted after each comment.

*1. Comment: The manuscript presents an interesting study on the estimation of tropical cyclone wind hazards. The topic falls in the scope of Natural Hazards and Earth System Sciences (NHESS). Generally, the paper is well written and organized. Some new findings different from suggestions in current specifications are highlighted and discussed. The presented research is of great importance to the wind-resistant design in coastal areas of China. The manuscript can be accepted for publication after minor revisions.*

**Response:** We really thanks for your careful review and valuable suggestions. We agree with all your comments and we have revised the manuscript accordingly.

*2. Comment: The values of the shape parameter of radial pressure profile in Fig. 11. Holland (1980) suggested that it should fall in the range [1.0, 2.5]. Vickery et al. (2000) suggested the range should be [0.5, 2.5]. There are a number of points larger than 2.5 in Fig. 11, which goes against our conventional cognition. Please give some essential explanations to clarify this point. i) Holland, G. J.: An analytic model of the wind and pressure profiles in hurricanes, Monthly Weather Review, 108, 1212-1218, 1980. ii) Vickery, P. J., Skerlj, P. F., Steckley, A. C., and Twisdale, L. A.: Hurricane Wind Field Model for Use in Hurricane Simulations, Journal of Structural Engineering, 126, 1203-1221, 2000.*

**Response:** Thanks for your comment. The difference is mainly attributed to the use of different wind field models and data sources. As listed in Table 1, the pressure and wind speed data sources were commonly employed to extract the $R_{max}$ and $B$ using different fitting models.

Table 1 Use of data source and fitting model for $R_{max}$ and $B$

| Data source | Fitting model | Reference |
|---|---|---|
| Surface pressure | Holland pressure model | Holland, 1980; Zhao et al., 2013; Fang et al., 2018b |
| Surface wind speed | Gradient and boundary layer wind models | Vickery et al., 2008; Fang et al., 2019; Zhao et al., 2020 |
| Upper level pressure | Convert to surface pressure | Vickery et al., 2000, 2008 |
| Upper level wind speed | Gradient wind model | Vickery et al., 2000 |

**Holland pressure model:**

$$P_{rs} = P_{cs} + \Delta P_s \cdot \exp\left[-\left(\frac{R_{max,s}}{r}\right)^{B_s}\right] \tag{1}$$

in which subscripts $s$ and $r$ denote surface values at the radius of $r$, $P_{rs}$= surface air pressure at radius of $r$ from the typhoon's axis (hPa), $P_{cs}$ = central pressure (hPa), $\Delta P_s = P_{ns} - P_{cs}$ is the central pressure difference (hPa).

**Gradient wind model:**

$$V_g = \frac{V_{T\theta} - fr}{2} + \sqrt{\left(\frac{V_{T\theta} - fr}{2}\right)^2 + \frac{r}{\rho_g}\frac{\partial P_g}{\partial r}} \tag{2}$$

in which $V_{T\theta} = -V_T \cdot sin(\theta - \theta_T)$, $V_T$ is the translation speed $(m/s)$, $\theta_T$ and $\theta$ are the translation direction and the direction of interest (counterclockwise positive from the east, °), $f$ is the Coriolis force, $\rho_g$ $(kg/m^3)$ and $P_g$ $(hPa)$ are the air density and pressure at gradient layer.

The pressure data (direct surface observations or converted from upper-level observations) can be directly applied to Eq. (1) to obtain $R_{max,s}$ and $B_s$, which is considered as the most physically reasonable method. Vickery et al. (2000, 2008) utilized the surface pressures converted from flight-level reconnaissance data to optimally obtain a pair of $R_{max,s}$ and $B_s$ for each traverse observation through the storm. Fang et al. (2018b) fitted the surface pressure data of landing typhoons observed by distributed meteorological stations in the mainland of China. However, when this equation is applied to model the wind speed field (assume $P_{rs} = P_g$) using Eq. (2) as used by most wind field models (Vickery et al., 2008), some inconsistencies could be introduced since the pressure distribution at free atmosphere is somewhat different from that at the surface. This can be approved from the results obtained by Willoughby et al (2004) and Vickery et al. (2000). Vickery et al. (2000) found that estimated $B$ from upper-level wind speed data using Eqs. (1)~(2) were about 20%~30% higher than that estimated from surface pressures. That means if Eq. (1) is estimated from the surface pressures, it cannot be directly applied to Eq. (2) due to the height-resolving characteristics of air density and pressures. And Eq. (2) is actually an approximate formula by neglecting the radial and vertical wind components. Moreover, even the pressure observation-based $R_{max,s}$ and $B_s$ were employed in the present wind field model, some inevitable errors on the estimations of wind speed would be introduced due to the simplification and linearization of the Navier-Stokes equations as discussed by Kepert and Wang (2001).

The other method is the use of wind speed observations. Vickery et al. (2008) used a boundary layer model to match the H* Wind surface wind field. The Holland pressure model, say Eq. (1) was also directly applied to Eq. (2) for calculating the gradient wind speed before converting to surface level. In fact, if Holland pressure model is considered to be valid at gradient level and substituted into Eq. (2), it is acceptable and self-consistent. That means $R_{max}$ and $B$ are estimated from gradient wind. And real wind field at gradient or surface level can be well captured although the real pressure field has a large deviation from Holland's model. The only problem is how to predetermine a gradient height since it is a variable and generally believed to increase from the storm center to

peripheral area.

Comparatively, the wind field model adopted in present study uses the surface level say 10 m above the ground as a standard height. The surface pressure was converted to gradient layer using a height-resolving pressure model (Fang et al., 2018a):

$$P_{rz} = \left\{ P_{cs} + \Delta P_s \cdot exp \left[ -\left( \frac{R_{max,s}}{r} \right)^{B_s} \right] \right\} \cdot \left( 1 - \frac{gkz}{R_d \theta_v} \right)^{\frac{1}{k}} \tag{3}$$

Then, an analytical boundary layer wind field model was utilized to calculate the surface wind speed (Fang et al., 2018a). The maximum gradient wind speed is considered to be positively correlated with the central pressure difference and $B_s$. To fit a specific real wind speed, a higher value of $B_s$ is required due to the decrease of central pressure difference from the surface to gradient layer when compared to no consideration of height-resolving characteristics of pressure field. Moreover, the analytical boundary layer model disregards some nonlinear terms and neglects the non-axisymmetric effects (Fang et al., 2018a), a larger $B_s$ is usually fitted to compensate for the deficiency of the model.

It is noteworthy that the surface pressures modeled by Eq. (1) using the fitting pair of $R_{max,s}$ and $B_s$ in this study could have a remarkable difference from the real pressures, but the modeled wind field is forced to match the observations as closely as possible to increase the accuracy of wind hazards estimation. More details regarding the extraction of $R_{max,s}$ and $B_s$ used in this study have been discussed in another study and in review (Zhao et al., 2020).

Explanations were also added in the revised manuscript in Lines 219-224 as:

"It is noteworthy that the fitted values of $B_s$ are slightly higher than traditional results, i.e. Vickery et al. (2000b, 2008) while $R_{max,s}$ are almost unchanged. This is mainly attributed to the use of surface wind data and an analytical wind field model in this study (Fang et al., 2018a, 2019b). To fit a specific real wind speed, a higher value of $B_s$ is required due to the decrease of central pressure difference from the surface to gradient layer when compared to no consideration of height-resolving characteristics of pressure field. Moreover, the analytical boundary layer model disregards some nonlinear terms and neglects the non-axisymmetric effects (Fang et al., 2018a), a larger $B_s$ is usually fitted to compensate for the deficiency of the model."

Reference

Holland, G. J.: An analytic model of the wind and pressure profiles in hurricanes, Monthly Weather Review, 108, 1212-1218, 1980.

Fang, G., Zhao, L., Cao, S., Ge, Y., and Pang W.: A novel analytical model for wind field simulation under typhoon boundary layer considering multi-field correlation and height-dependency, Journal of Wind Engineering and Industrial Aerodynamics, 175, 77-89, 2018a.

Fang G, Zhao L, Song L, et al. Reconstruction of radial parametric pressure field near ground surface of landing typhoons in Northwest Pacific Ocean[J]. Journal of Wind Engineering and Industrial Aerodynamics, 2018b, 183:223-234.

Fang, G., Pang, W., Zhao, L., Cao, S., and Ge, Y.: Towards a refined estimation of typhoon wind hazards: Parametric modelling and upstream terrain effects, The 15th International Conference on Wind Engineering, Beijing, China; September 1-6, 2019b.

Kepert J, Wang Y. The dynamics of boundary layer jets within the tropical cyclone core. Part II: Nonlinear enhancement. Journal of the atmospheric sciences, 2001, 58 (17), 2485-2501

Vickery P J, Skerlj P F, Steckley A C, et al. Hurricane Wind Field Model for Use in Hurricane Simulations[J]. Journal of Structural Engineering, 2000, 126(10):1203-1221.

Vickery P J , Wadhera D . Statistical Models of Holland Pressure Profile Parameter and Radius to Maximum Winds of Hurricanes from Flight-Level Pressure and H*Wind Data[J]. Journal of Applied Meteorology and Climatology, 2008, 47(10):2497-2517.

Willoughby H E , Rahn M E . Parametric Representation of the Primary Hurricane Vortex. Part Ⅰ: Observations and Evaluation of the Holland (1980) Model[J]. Monthly Weather Review, 2004, 132(12):p.3033-3048.

Zhao L , Lu A , Zhu L , et al. Radial pressure profile of typhoon field near ground surface observed by distributed meteorologic stations[J]. Journal of Wind Engineering and Industrial Aerodynamics, 2013, 122:105-112.

Zhao L., Fang G. S., Pang W., Rawal P., Cao S. Y., and Ge Y. J.. Toward a refined estimation of typhoon wind hazards: Parametric modeling and upstream terrain effects, Journal of Wind Engineering & Industrial Aerodynamics, 2020. (in review).

*3. Comment:* Fig. 11 can be improved to avoid some data points obscured by legend.

**Response:** Thanks for your careful reading and comments. Fig.11 has been replotted as follows.

[Figure]

**Figure 11: Comparison of $B_s$ between model and real observations: (a~d) relations between $B_s(i)$, $B_s(i-1)$, $lnR_{max,s}(i+1)$, $\Delta P(i+1)$ and $B_s(i+1)$ without errors; (e~h) relations between $B_s(i)$, $B_s(i-1)$, $lnR_{max,s}(i+1)$, $\Delta P(i+1)$ and $B_s(i+1)$ with errors ($\rho_{real}$ is the correlation coefficient for real observation data)**

*4. Comment:* Lines 24, 37, 40, 416, 440: characterizing tropical cyclone as 'non-synoptic' is questionable. Tropical cyclone is actually a non-frontal synoptic-scale cyclone as discussed by Vallis et al (2019). Vallis, M. B., Loredo-Souza, A. M., Ferreira, V., Nascimento E. L.: Classification and identification of synoptic and non-synoptic extreme

wind events from surface observations in South America, Journal of Wind Engineering and Industrial Aerodynamics, 193, 2019, 103963.

**Response:** We really appreciate you for pointing out the misunderstanding of the concepts. We carefully examine the concept of synoptic scale winds and tropical cyclone. As explained by National Oceanic and Atmospheric Administration (NOAA) (https://www.nhc.noaa.gov/aboutgloss.shtml) "tropical cyclone is a warm-core non-frontal synoptic-scale cyclone, originating over tropical or subtropical waters, with organized deep convection and a closed surface wind circulation about a well-defined center". Vallis et al (2019) characterized the extreme wind events into synoptic, non-synoptic and tropical cyclone (TC) events. The word "synoptic" has been replaced by the "non-TC" in the revised manuscript.

*5. Comment:* Although this paper focuses on the characteristics of the mean components of tropical cyclones, some discussions on the fluctuation components (stationary or nonstationary) are suggested to be supplemented in the introduction part. The following references may do some help. i) Modelling of longitudinal evolutionary power spectral density of typhoon winds considering high-frequency subrange. Journal of Wind Engineering and Industrial Aerodynamics 2019, 193, 103957. ii) Reduced-Hermite bifoldinterpolation assisted schemes for the simulation of random wind field. Probabilistic Engineering Mechanics 2018, 53, 126-142.

**Response:** Thanks for your recommendation. Authors have carefully read suggested papers and found their great contributions to understand the fluctuating characteristics of TC winds. They provide us with a lot of information to further simulate the fluctuating components of TC winds in the future. They have also been added to our reference.

*6. Comment:* There are some typos in the manuscript, e.g., In line 124, "influence" should be "influence"; In line 149, "modeling" was used while "modelling" was utilized in line 154. Please use a consistent form.

**Response:** Thanks for your careful reading and comments. The correction has been made. And similar typos have been carefully checked and revised.

---

## Author Comment (AC2) · 12 Apr 2020

Dear Dr. Huang

We would like to thank you for your careful and thorough reading of our manuscript and for the thoughtful comments and constructive suggestions. Your comments are of great help to improve the quality of this manuscript. We agree with all your comments and we have revised the manuscript accordingly. We are already crafting a revised version of the paper. Attached please find our responses.

Regards Authors

Please also note the supplement to this comment:

[Figure]

https://www.nat-hazards-earth-syst-sci-discuss.net/nhess-2019-375/nhess-2019-375-AC2-supplement.pdf

[Figure]

**Supplement:**

**Responses to *Dr. Huang's Comments**

**Manuscript Number: nhess-2019-375**

**Title of Paper:** Estimation of Tropical Cyclone Wind Hazards in Coastal Regions of China

**Journal:** Natural Hazards and Earth System Sciences (NHESS)

Dear Dr. Huang

We would like to thank you for your careful and thorough reading of our manuscript and for the thoughtful comments and constructive suggestions. Your comments are of great help to improve the quality of this manuscript. We agree with all your comments and we have revised the manuscript accordingly. We are already crafting a revised version of the paper.

*1. Comment: This manuscript estimates the tropical cyclone wind hazards in southeastern coastal region of China. Two typhoon wind field parameters, i.e. radius to maximum winds $R_{max,s}$ and shape parameter of radial pressure profile $B_s$ are identified using JMA best track dataset coupled with a boundary layer wind field model. TC wind hazard curves in terms of design wind speed versus return periods for major coastal cities of China are developed. The topic of this study is in-line with the journal of "Natural Hazards and Earth System Sciences (NHESS)". Generally, the paper is a well-organized study and worth to be published. The obtained results will be valuable to the researchers and engineers in this field.*

**Response:** We really appreciate your positive feedback. We agree with all your comments and we have revised the manuscript accordingly.

*2. Comment: The major concern is the use of the wind-driven $R_{max,s}$ and $B_s$. The results in Figs. 11 and 13 show that $B_s$ and $R_{max,s}$ have a positive correlation which is inconsistent with the findings by Vickery et al. (2008). And few values of $B_s$ are higher than 2.5 which fall outside the range of 0.5~2.5 suggested by Vickery et al. (2000). Please explain.*

*Vickery, P. J., Skerlj, P. F., Steckley, A. C., and Twisdale, L. A.: Hurricane Wind Field Model for Use in Hurricane Simulations, Journal of Structural Engineering, 126, 1203-1221, 2000.*

*Vickery, P. J. and Wadhera, D.: Statistical Models of Holland Pressure Profile Parameter and Radius to Maximum Winds of Hurricanes from Flight-Level Pressure and H\*Wind Data, Journal of Applied Meteorology and Climatology, 47, 2497-2517, 2008.*

**Response:** Thanks for your comment. Regarding the value difference of $B_s$ identified in this study, similar response replied to Anonymous Referee #1 was present as follow:

The difference is mainly attributed to the use of different wind field models and data sources. As listed in Table 1, the pressure and wind speed data sources were commonly employed to extract the $R_{max}$ and $B$ using different fitting models.

Table 1 Use of data source and fitting model for $R_{max}$ and $B$

| Data source | Fitting model | Reference |
|---|---|---|
| Surface pressure | Holland pressure model | Holland, 1980; Zhao et al., 2013; Fang et al., 2018b |
| Surface wind speed | Gradient and boundary layer wind models | Vickery et al., 2008; Fang et al., 2019; Zhao et al., 2020 |
| Upper level pressure | Convert to surface pressure | Vickery et al., 2000, 2008 |
| Upper level wind speed | Gradient wind model | Vickery et al., 2000 |

**Holland pressure model:**

$$P_{rs} = P_{cs} + \Delta P_s \cdot \exp\left[-\left(\frac{R_{max,s}}{r}\right)^{B_s}\right] \tag{1}$$

in which subscripts $s$ and $r$ denote surface values at the radius of $r$, $P_{rs}$ = surface air pressure at radius of $r$ from the typhoon's axis (hPa), $P_{cs}$ = central pressure (hPa), $\Delta P_s = P_{ns} - P_{cs}$ is the central pressure difference (hPa).

**Gradient wind model:**

$$V_g = \frac{V_{T\theta} - fr}{2} + \sqrt{\left(\frac{V_{T\theta} - fr}{2}\right)^2 + \frac{r}{\rho_g}\frac{\partial P_g}{\partial r}} \tag{2}$$

in which $V_{T\theta} = -V_T \cdot sin(\theta - \theta_T)$, $V_T$ is the translation speed $(m/s)$, $\theta_T$ and $\theta$ are the translation direction and the direction of interest (counterclockwise positive from the east, °), $f$ is the Coriolis force, $\rho_g$ $(kg/m^3)$ and $P_g$ $(hPa)$ are the air density and pressure at gradient layer.

The pressure data (direct surface observations or converted from upper-level observations) can be directly applied to Eq. (1) to obtain $R_{max,s}$ and $B_s$, which is considered as the most physically reasonable method. Vickery et al. (2000, 2008) utilized the surface pressures converted from flight-level reconnaissance data to optimally obtain a pair of $R_{max,s}$ and $B_s$ for each traverse observation through the storm. Fang et al. (2018b) fitted the surface pressure data of landing typhoons observed by distributed meteorological stations in the mainland of China. However, when this equation is applied to model the wind speed field (assume $P_{rs} = P_g$) using Eq. (2) as used by most wind field models (Vickery et al., 2008), some inconsistencies could be introduced since the pressure distribution at free atmosphere is somewhat different from that at the surface. This can be approved from the results obtained by Willoughby et al (2004) and Vickery et al. (2000). Vickery et al. (2000) found that estimated $B$ from upper-level wind speed data using Eqs. (1)~(2) were about 20%~30% higher than that estimated from surface pressures. That means if Eq. (1) is estimated from the surface pressures, it cannot be directly applied to Eq. (2) due to the height-resolving characteristics of air density and pressures. And Eq. (2) is actually an approximate formula by neglecting the radial and vertical wind components. Moreover, even the pressure observation-based $R_{max,s}$ and $B_s$ were employed in the present wind field model, some inevitable errors on the estimations of wind speed would be introduced due to the simplification and linearization of the Navier-Stokes equations as discussed by Kepert and

Wang (2001).

The other method is the use of wind speed observations. Vickery et al. (2008) used a boundary layer model to match the H* Wind surface wind field. The Holland pressure model, say Eq. (1) was also directly applied to Eq. (2) for calculating the gradient wind speed before converting to surface level. In fact, if Holland pressure model is considered to be valid at gradient level and substituted into Eq. (2), it is acceptable and self-consistent. That means $R_{max}$ and $B$ are estimated from gradient wind. And real wind field at gradient or surface level can be well captured although the real pressure field has a large deviation from Holland's model. The only problem is how to predetermine a gradient height since it is a variable and generally believed to increase from the storm center to peripheral area.

Comparatively, the wind field model adopted in present study uses the surface level say 10 m above the ground as a standard height. The surface pressure was converted to gradient layer using a height-resolving pressure model (Fang et al., 2018a):

$$P_{rz} = \left\{ P_{cs} + \Delta P_s \cdot exp\left[-\left(\frac{R_{max,s}}{r}\right)^{B_s}\right] \right\} \cdot \left(1 - \frac{gkz}{R_d \theta_v}\right)^{\frac{1}{k}} \tag{3}$$

Then, an analytical boundary layer wind field model was utilized to calculate the surface wind speed (Fang et al., 2018a). The maximum gradient wind speed is considered to be positively correlated with the central pressure difference and $B_s$. To fit a specific real wind speed, a higher value of $B_s$ is required due to the decrease of central pressure difference from the surface to gradient layer when compared to no consideration of height-resolving characteristics of pressure field. Moreover, the analytical boundary layer model disregards some nonlinear terms and neglects the non-axisymmetric effects (Fang et al., 2018a), a larger $B_s$ is usually fitted to compensate for the deficiency of the model.

It is noteworthy that the surface pressures modeled by Eq. (1) using the fitting pair of $R_{max,s}$ and $B_s$ in this study could have a remarkable difference from the real pressures, but the modeled wind field is forced to match the observations as closely as possible to increase the accuracy of wind hazards estimation. More details regarding the extraction of $R_{max,s}$ and $B_s$ used in this study have been discussed in another study and in review (Zhao et al., 2020).

Explanations were also added in the revised manuscript in Lines 219-224 as:

"It is noteworthy that the fitted values of $B_s$ are slightly higher than traditional results, i.e. Vickery et al. (2000b, 2008) while $R_{max,s}$ are almost unchanged. This is mainly attributed to the use of surface wind data and an analytical wind field model in this study (Fang et al., 2018a, 2019b). To fit a specific real wind speed, a higher value of $B_s$ is required due to the decrease of central pressure difference from the surface to gradient layer when compared to no consideration of height-resolving characteristics of pressure field. Moreover, the analytical boundary layer model disregards some nonlinear terms and neglects the non-axisymmetric effects (Fang et al., 2018a), a larger $B_s$ is usually fitted to compensate for the deficiency of the model."

The correlation between $B_s$ and $R_{max,s}$ is positive in this study while negative correlation was found by Vickery

et al. (2008). This could attribute to the difference of TC structure in Western Pacific and Atlantic Ocean. The difference of best track dataset as well as the use of different fitting methods could also be responsible for this difference. Polamuri (2019) also found a positive correlation between $B_s$ and $R_{max,s}$ when JMA best track dataset was utilized.

Reference

Holland, G. J.: An analytic model of the wind and pressure profiles in hurricanes, Monthly Weather Review, 108, 1212-1218, 1980.

Fang, G., Zhao, L., Cao, S., Ge, Y., and Pang W.: A novel analytical model for wind field simulation under typhoon boundary layer considering multi-field correlation and height-dependency, Journal of Wind Engineering and Industrial Aerodynamics, 175, 77-89, 2018a.

Fang G, Zhao L, Song L, et al. Reconstruction of radial parametric pressure field near ground surface of landing typhoons in Northwest Pacific Ocean[J]. Journal of Wind Engineering and Industrial Aerodynamics, 2018b, 183:223-234.

Fang, G., Pang, W., Zhao, L., Cao, S., and Ge, Y.: Towards a refined estimation of typhoon wind hazards: Parametric modelling and upstream terrain effects, The 15th International Conference on Wind Engineering, Beijing, China; September 1-6, 2019b.

Kepert J, Wang Y. The dynamics of boundary layer jets within the tropical cyclone core. Part II: Nonlinear enhancement. Journal of the atmospheric sciences, 2001, 58 (17), 2485-2501

Polamuri S H, 2019. Projections of typhoon wind speeds under climate change in Asia Pacific Basin, Ph.D. Thesis, Glenn Department of Civil Engineering, Clemson University, South Carolina, United States.

Vickery P J, Skerlj P F, Steckley A C, et al. Hurricane Wind Field Model for Use in Hurricane Simulations[J]. Journal of Structural Engineering, 2000, 126(10):1203-1221.

Vickery P J , Wadhera D . Statistical Models of Holland Pressure Profile Parameter and Radius to Maximum Winds of Hurricanes from Flight-Level Pressure and H*Wind Data[J]. Journal of Applied Meteorology and Climatology, 2008, 47(10):2497-2517.

Willoughby H E , Rahn M E . Parametric Representation of the Primary Hurricane Vortex. Part Ⅰ: Observations and Evaluation of the Holland (1980) Model[J]. Monthly Weather Review, 2004, 132(12):p.3033-3048.

Zhao L , Lu A , Zhu L , et al. Radial pressure profile of typhoon field near ground surface observed by distributed meteorologic stations[J]. Journal of Wind Engineering and Industrial Aerodynamics, 2013, 122:105-112.

Zhao L., Fang G. S., Pang W., Rawal P., Cao S. Y., and Ge Y. J.. Toward a refined estimation of typhoon wind hazards: Parametric modeling and upstream terrain effects, Journal of Wind Engineering & Industrial Aerodynamics, 2020. (in review).

***3. Comment:*** *The titles of section 2.1 and 2.2 are identical. Please check.*

**Response:** Thanks for your comment. Section 2.2 should be "Statistical correlations". The correction has been made.

***4. Comment:*** *Line 409, "…show satisfactory agreement with…", consider use "…show a satisfactory agreement with…" or "…are in satisfactory agreement with…".*

**Response:** Thanks for your careful reading. The correction has been made.

**5. Comment:** *A similar study performed by Wu and Huang (2019) is suggested to be compared and discussed. Wu F., and Huang G.: Refined Empirical Model of Typhoon Wind Field and Its Application in China, Journal of Structural Engineering, 145(11): 04019122, 2019.*

**Response:** Thanks for your recommendation. Authors have carefully read the suggested paper. It provides us with a lot of information to further understand the typhoon hazard in coastal regions of China. They have also been added to our reference. It was also compared with present and other studies in Lines 368 and 408.

"…A similar trend can also be observed from the differences between Li and Hong (2016), Chen and Duan (2017), Wu and Hung (2019) and the codes…"

"…The wind speeds predicted by Wu and Huang (2019) are similar to those estimated by Li and Hong (2016) which mainly attributes to the use of the same best track dataset as well as $R_{max}$ and $B$ models…"

---

## Author Comment (AC3) · 12 Apr 2020

Dear Dr. He, We would like to thank you for your careful and thorough reading of our manuscript and for the thoughtful comments and constructive suggestions. We are already crafting a revised version of the paper. Please, find below the referees' comments repeated in italics and our responses inserted after each comment. Regards, Authors

Please also note the supplement to this comment:
https://www.nat-hazards-earth-syst-sci-discuss.net/nhess-2019-375/nhess-2019-375-AC3-supplement.pdf

[Figure]

**Supplement:**

**Responses to *Dr. He**

**Manuscript Number: nhess-2019-375**

**Title of Paper:** Estimation of Tropical Cyclone Wind Hazards in Coastal Regions of China

**Journal:** Natural Hazards and Earth System Sciences (NHESS)

Dear Dr. He,

We would like to thank you for your careful and thorough reading of our manuscript and for the thoughtful comments and constructive suggestions. We are already crafting a revised version of the paper. Please, find below the referees' comments repeated in italics and our responses inserted after each comment.

*1. Comment: This article presents a detailed study on the estimation of TC-wind hazards in southeast coast of China. Values of key parameters of TCs, i.e., RMW and Holland-B, are firstly estimated by fitting TC best-track records from JMA via a TC wind field model. These results are then utilized to generate a number of recursive models for corresponding parameters of TC activities and TC wind field. The proposed recursive models are further exploited in conjunction of the TC wind field model to estimate TC extreme winds associated with different return periods at several selected coastal cities. Finally, results of TC wind hazards obtained from this study are compared with those stipulated in codes or the ones documented by peers. Overall, this work is well written and the analysis process is scrupulous, which makes the findings convincing. It is expected that the findings can provide further insights to better understand the design speeds at coastal areas of China. This reviewer actually has few specific comments for the improvement of this article, but there are still some issues that should be clarified.*

**Response:** We really appreciate your positive feedback and your valuable suggestions. We agree with all your comments and we have revised the manuscript accordingly.

*2. Comment: RMW and Holland-B are two key parameters whose values can influence the simulation results of TC wind field severely. Actually, some researchers pointed out that the majority of uncertainty for assessing TC wind hazards should be attributed to the estimation of these parameters. In this regard, great efforts are encouraged to pay to accurate estimation of their values. Basically, there are two kinds of methods which are driven by wind speed records and pressure records, respectively. According to the pioneering work by Holland (1980), RMW and Holland-B are defined under the context of TC pressure field, which potentially indicates that the pressure-data driven method is more straightforward, and possibly more effective. As stated in my general comments, the authors choose the speed-data driven method. Besides the above consideration, there are also several uncertainty sources: (1) even though the authors explain much for choosing TC records from JMA, the basic records herein still belong to the "best-track" data, which means they may differ from the real noticeably. (2) TC wind field possesses asymmetric*

features, while according to the statements in this study, the best-track information for estimating these two parameters may practically account for symmetric TC wind field. If this is the case, the estimation accuracy could be degraded. (3) The authors use a height-resolving model to depict TC wind field, while the best-track TC information is given at a fixed level. Please detail in the context how to deal with the inconsistency in terms of height level between model and dataset (including what altitude should the best-track data best account for). It is also suggested that the obtained values of RMW and Holland-B be statistically compared with their counterparts in previous studies.

**Response:** Thanks for your comment. Indeed, as you mentioned, $R_{max}$ and $B$ have significant effects on the estimation of TC wind hazards. As replied to Anonymous Referee #1 and Dr. Huang, Table 1 lists the fitting methods for $R_{max}$ and $B$. The pressure and wind speed data sources were commonly employed to extract the $R_{max}$ and $B$ using different fitting models.

Table 1 Use of data source and fitting model for $R_{max}$ and $B$

| Data source | Fitting model | Reference |
|---|---|---|
| Surface pressure | Holland pressure model | Holland, 1980; Zhao et al., 2013; Fang et al., 2018b |
| Surface wind speed | Gradient and boundary layer wind models | Vickery et al., 2008; Fang et al., 2019; Zhao et al., 2020 |
| Upper level pressure | Convert to surface pressure | Vickery et al., 2000, 2008 |
| Upper level wind speed | Gradient wind model | Vickery et al., 2000 |

**Holland pressure model:**

$$P_{rs} = P_{cs} + \Delta P_s \cdot \exp\left[-\left(\frac{R_{max,s}}{r}\right)^{B_s}\right] \tag{1}$$

in which subscripts $s$ and $r$ denote surface values at the radius of $r$, $P_{rs}$= surface air pressure at radius of $r$ from the typhoon's axis (hPa), $P_{cs}$ = central pressure (hPa), $\Delta P_s = P_{ns} - P_{cs}$ is the central pressure difference (hPa).

**Gradient wind model:**

$$V_g = \frac{V_{T\theta} - fr}{2} + \sqrt{\left(\frac{V_{T\theta} - fr}{2}\right)^2 + \frac{r}{\rho_g}\frac{\partial P_g}{\partial r}} \tag{2}$$

in which $V_{T\theta} = -V_T \cdot sin(\theta - \theta_T)$, $V_T$ is the translation speed $(m/s)$, $\theta_T$ and $\theta$ are the translation direction and the direction of interest (counterclockwise positive from the east, °), $f$ is the Coriolis force, $\rho_g$ $(kg/m^3)$ and $P_g$ $(hPa)$ are the air density and pressure at gradient layer.

The pressure data (direct surface observations or converted from upper-level observations) can be directly applied to Eq. (1) to obtain $R_{max,s}$ and $B_s$, which is considered as the most physically reasonable method. Vickery et al. (2000, 2008) utilized the surface pressures converted from flight-level reconnaissance data to optimally obtain a pair of $R_{max,s}$ and $B_s$ for each traverse observation through the storm. Fang et al. (2018b) fitted the surface pressure data of landing typhoons observed by distributed meteorological stations in the mainland of China. However, when this equation is applied to model the wind speed field (assume $P_{rs} = P_g$) using Eq. (2) as used by

most wind field models (Vickery et al., 2008), some inconsistencies could be introduced since the pressure distribution at free atmosphere is somewhat different from that at the surface. This can be approved from the results obtained by Willoughby et al (2004) and Vickery et al. (2000). Vickery et al. (2000) found that estimated $B$ from upper-level wind speed data using Eqs. (1)~(2) were about 20%~30% higher than that estimated from surface pressures. That means if Eq. (1) is estimated from the surface pressures, it cannot be directly applied to Eq. (2) due to the height-resolving characteristics of air density and pressures. And Eq. (2) is actually an approximate formula by neglecting the radial and vertical wind components. Moreover, even the pressure observation-based $R_{max,s}$ and $B_s$ were employed in the present wind field model, some inevitable errors on the estimations of wind speed would be introduced due to the simplification and linearization of the Navier-Stokes equations as discussed by Kepert and Wang (2001).

The other method is the use of wind speed observations. Vickery et al. (2008) used a boundary layer model to match the H* Wind surface wind field. The Holland pressure model, say Eq. (1) was also directly applied to Eq. (2) for calculating the gradient wind speed before converting to surface level. In fact, if Holland pressure model is considered to be valid at gradient level and substituted into Eq. (2), it is acceptable and self-consistent. That means $R_{max}$ and $B$ are estimated from gradient wind. And real wind field at gradient or surface level can be well captured although the real pressure field has a large deviation from Holland's model. The only problem is how to predetermine a gradient height since it is a variable and generally believed to increase from the storm center to peripheral area.

Comparatively, the wind field model adopted in present study uses the surface level say 10 m above the ground as a standard height. The surface pressure was converted to gradient layer using a height-resolving pressure model (Fang et al., 2018a):

$$P_{rz} = \left\{ P_{cs} + \Delta P_s \cdot exp\left[ -\left( \frac{R_{max,s}}{r} \right)^{B_s} \right] \right\} \cdot \left( 1 - \frac{gkz}{R_d \theta_v} \right)^{\frac{1}{k}} \tag{3}$$

Then, an analytical boundary layer wind field model was utilized to calculate the surface wind speed (Fang et al., 2018a). The maximum gradient wind speed is considered to be positively correlated with the central pressure difference and $B_s$. To fit a specific real wind speed, a higher value of $B_s$ is required due to the decrease of central pressure difference from the surface to gradient layer when compared to no consideration of height-resolving characteristics of pressure field. Moreover, the analytical boundary layer model disregards some nonlinear terms and neglects the non-axisymmetric effects (Fang et al., 2018a), a larger $B_s$ is usually fitted to compensate for the deficiency of the model.

It is noteworthy that the surface pressures modeled by Eq. (1) using the fitting pair of $R_{max,s}$ and $B_s$ in this study could have a remarkable difference from the real pressures, but the modeled wind field is forced to match the observations (wind speed information in best track dataset ) as closely as possible to increase the accuracy of wind hazards estimation. More details regarding the extraction of $R_{max,s}$ and $B_s$ used in this study have been discussed in another study and in review (Zhao et al., 2020).

As stated in Line 90, the surface wind speed information is provided, say at height of 10 m. The height-resolving TC boundary layer wind field model employed in this study allows to reproduce the wind field at any given height. So $R_{max,s}$ and $B_s$ were all fitted at a height of 10 m.

Reference

Holland, G. J.: An analytic model of the wind and pressure profiles in hurricanes, Monthly Weather Review, 108, 1212-1218, 1980.

Fang, G., Zhao, L., Cao, S., Ge, Y., and Pang W.: A novel analytical model for wind field simulation under typhoon boundary layer considering multi-field correlation and height-dependency, Journal of Wind Engineering and Industrial Aerodynamics, 175, 77-89, 2018a.

Fang G, Zhao L, Song L, et al. Reconstruction of radial parametric pressure field near ground surface of landing typhoons in Northwest Pacific Ocean[J]. Journal of Wind Engineering and Industrial Aerodynamics, 2018b, 183:223-234.

Fang, G., Pang, W., Zhao, L., Cao, S., and Ge, Y.: Towards a refined estimation of typhoon wind hazards: Parametric modelling and upstream terrain effects, The 15th International Conference on Wind Engineering, Beijing, China; September 1-6, 2019b.

Kepert J, Wang Y. The dynamics of boundary layer jets within the tropical cyclone core. Part II: Nonlinear enhancement. Journal of the atmospheric sciences, 2001, 58 (17), 2485-2501

Vickery P J, Skerlj P F, Steckley A C, et al. Hurricane Wind Field Model for Use in Hurricane Simulations[J]. Journal of Structural Engineering, 2000, 126(10):1203-1221.

Vickery P J , Wadhera D . Statistical Models of Holland Pressure Profile Parameter and Radius to Maximum Winds of Hurricanes from Flight-Level Pressure and H*Wind Data[J]. Journal of Applied Meteorology and Climatology, 2008, 47(10):2497-2517.

Willoughby H E , Rahn M E . Parametric Representation of the Primary Hurricane Vortex. Part Ⅰ: Observations and Evaluation of the Holland (1980) Model[J]. Monthly Weather Review, 2004, 132(12):p.3033-3048.

Zhao L , Lu A , Zhu L , et al. Radial pressure profile of typhoon field near ground surface observed by distributed meteorologic stations[J]. Journal of Wind Engineering and Industrial Aerodynamics, 2013, 122:105-112.

Zhao L., Fang G. S., Pang W., Rawal P., Cao S. Y., and Ge Y. J.. Toward a refined estimation of typhoon wind hazards: Parametric modeling and upstream terrain effects, Journal of Wind Engineering & Industrial Aerodynamics, 2020. (in review).

***3. Comment:*** Why do the authors choose a height-resolving TC wind field model rather than others, e.g., a slab model, in this study? To match it with the best-track data which account for a height beyond near ground range? Please clarify.

**Response:** Thanks for your comment. JMA best track dataset provides the surface wind speed information (at height of 10 m). To fit the $R_{max,s}$ and $B_s$, the TC boundary layer wind field model should be able to reproduce the surface wind field. The height-resolving boundary layer wind field model developed by Meng et al. (1995) and enhanced by Fang et al. (2018a) is adopted in this study. The slab model usually defines the gradient height as a constant value. The surface wind speed is estimated by an empirically based reduction relationship between the gradient and the

near ground wind velocity. The accuracy of the slab model, especially for simulating the typhoon boundary layer, is not well-behaved because it relies heavily on modification from observation data and empirical analysis. Furthermore, the spatial velocity distribution in the typhoon boundary layer and the terrain effects are ignored to some extent. Comparatively, the height-resolving wind field model is an improved method for directly solving the Navier-Stokes equation and is based on several simplified semi-analytical algorithms. The features of the wind field can be described approximately and the terrain types, treated as roughness-related parameters, are included in the updated wind field model.

As stated in Line 183, $R_{max,s}$ and $B_s$ were fitted at surface level.

"A height-resolving TC boundary layer model developed by Meng et al. (1995) and enhanced by Fang et al. (2018a) is adopted in this study. It is also used to extract two typical TC wind field parameters: radius to maximum wind speed ($R_{max,s}$) and radial pressure profile shape parameter ($B_s$) at surface level."

Reference

Meng, Y., Matsui, M., Hibi, K.: An analytical model for simulation of the wind field in a typhoon boundary layer, Journal of Wind Engineering and Industrial Aerodynamics, 56, 291-310, 1995.

Fang, G., Zhao, L., Cao, S., Ge, Y., and Pang W.: A novel analytical model for wind field simulation under typhoon boundary layer considering multi-field correlation and height-dependency, Journal of Wind Engineering and Industrial Aerodynamics, 175, 77-89, 2018a.

*4. **Comment:*** Another comment is about the gradient height. It is assumed in this study that the gradient height is equal to 500 m. However, observational results show that TC depth tends to deepen when TCs get close to coastal areas. Will the inaccuracy of TC depth influence the estimation results? If so, to what an extent?

**Response:** We really appreciate you for pointing out this. We assumed the gradient height of 500 m only when we roughly converted the design wind speed suggested by Hong Kong Code (2004) to the wind speed associated with the reference exposure used in this study ($z_0 = 0.05$) for comparison purpose. As mentioned, observations show that the gradient height tends to increase when TCs get close to coastal areas. The height resolving boundary layer wind field model can reproduce the inner boundary layer of a TC at a given surface roughness length. For example, as shown in Fig.1, the vertical wind speed profiles of a synthetic TC are compared with that observed by dropsonde data (Giammanco et al., 2013). It can be noted that the wind field model well reproduces the vertical profiles. To predict the wind hazard curves at a specific site, a reference surface roughness length, say $z_0 = 0.05$ is employed. This is consistent with Chinese code. Moreover, the TC surface wind field can also be reproduced if the location-specific surface roughnesses are applied as studied by Fang et al. (2019) and Zhao et al. (2020). Fig. 2 shows an example of reproduced surface wind field of typhoon Rammasun at 06:00 UTC, 07/18, 2014 studied by Zhao et al. (2020).

[Figure]

Fig. 1 Comparison of vertical profiles between a synthetic TC and observations

[Figure]

Fig. 2 Wind field of strong typhoon Rammasun at 06:00 UTC, 07/18, 2014 (10 m): a) Wind field with a uniform $z_0$ (m/s); b) Directional $z_0$ (m); c) Wind field with directional $z_0$ (m/s); d) Elevation map (m); e) Directional $K_t$; f) Wind field with directional $z_0$ and $K_t$ (m/s);

Reference

Buildings Department, Hong Kong: Code of Practice on Wind Effects in Hong Kong 2004, The Government of the Hong Kong Special Administrative Region, 2004.

Buildings Department, Hong Kong: Explanatory Materials to the Code of Practice on Wind Effects in Hong Kong 2004, The Government of the Hong Kong Special Administrative Region, 2004.

Fang, G., Pang, W., Zhao, L., Cao, S., and Ge, Y.: Towards a refined estimation of typhoon wind hazards: Parametric modelling and upstream terrain effects, The 15th International Conference on Wind Engineering, Beijing, China; September 1-6, 2019b.

Zhao L., Fang G. S., Pang W., Rawal P., Cao S. Y., and Ge Y. J.. Toward a refined estimation of typhoon wind hazards: Parametric modeling and upstream terrain effects, Journal of Wind Engineering & Industrial Aerodynamics, 2020. (in review).

**_5. Comment:_** Some minor comments: 1) Line 21: under TC climates climate; 2) Lines 225-226: The critical value of K-S test (n = 161) is 0.1059 at a 5% significance level larger than the test statistics…

**Response:** Thanks for your careful reading and comments. The correction has been made. And similar typos have been carefully checked and revised.